# RADIO1D: Elastic Representations for Condensed Vision Modeling

Greg Heinrich [* 1]   Mike Ranzinger [* 1]   Collin McCarthy [* 1]   Natan Bagrov [1]   Eugene Khvedchenya [1]
Bryan Catanzaro [1]   Jan Kautz [1]   Andrew Tao [1]   Pavlo Molchanov [1]

## Abstract

This paper challenges the assumption that vision-language models (VLMs) require fixed patch-based 2D vision features. Analyzing fine-tuned vision encoders, we find that representations become increasingly abstract and less spatially coherent during VLM training. Notably, models trained with image-text alignment (such as SigLIP2) develop a small number of specialized tokens that effectively summarize global image content. Building on this, we introduce RADIO1D, which compresses images into a compact, variable-length 1D token sequence using multi-teacher knowledge distillation and an autoencoder design. The resulting representations exhibit strong hierarchical summarization, enabling accurate scene understanding–even with a single token–and support improved composition-aware image retrieval. In VLMs, RADIO1D provides flexible accuracy-efficiency tradeoffs through adjustable token counts, delivering competitive performance on diverse multimodal benchmarks with lower computational overhead and better accuracy.

## 1 Introduction

The advent of vision foundation models has revolutionized vision-language models (VLMs) by enabling efficient multimodal integration without extensive retraining. Flamingo pioneered this by leveraging frozen vision encoders like CLIP (Radford et al., 2021), resampling their features via a Perceiver module, and prefixing them into a frozen large language model for few-shot multimodal tasks such as visual question answering and captioning. Building on this, BLIP-2 (Li et al., 2023) introduced a lightweight Querying Transformer to bridge frozen CLIP-like vision encoders

*Equal contribution  [1]NVIDIA. Correspondence to: Greg Heinrich <gheinrich@nvidia.com>.

*Proceedings of the $43^{rd}$ International Conference on Machine Learning*, Seoul, South Korea. PMLR 306, 2026. Copyright 2026 by the author(s).

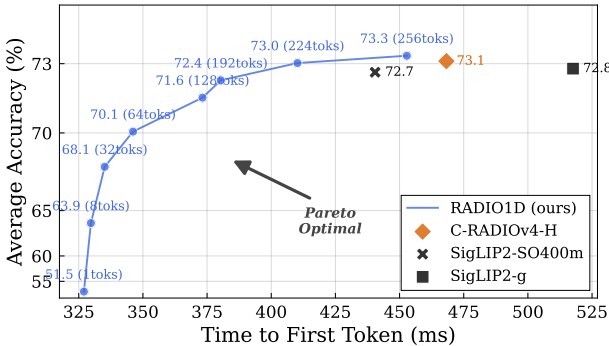

Average VLM Accuracy vs Time To First Token

*Figure 1.* Average accuracy across 10 VLM benchmarks. Vision Encoders (VE) are paired with a 9B Nemotron LLM and trained on 17M examples. All RADIO1D-based VLMs are obtained from the same initial VE checkpoint: RADIO1D enables flexible accuracy-latency trade-offs by varying the number of output tokens, unlike baselines with fixed outputs.

with frozen LLMs, achieving state-of-the-art zero-shot performance through two-stage pre-training. Kosmos-1 (Huang et al., 2023) further advanced the paradigm by projecting CLIP features and concatenating them with text tokens in a unified stream for perception-language alignment, supporting tasks like nonverbal reasoning. LLaVA (Liu et al., 2023) simplified this with direct linear projection of CLIP features into pre-trained LLMs, combined with visual instruction tuning using GPT-4-generated data, yielding versatile multimodal assistants.

Subsequent advancements saw SigLIP (Zhai et al., 2023) and its successor SigLIP2 (Tschannen et al., 2025) emerge as popular vision foundation models, offering improved efficiency and multilingual capabilities through sigmoid-based losses and enhanced pre-training recipes. Several top VLMs adopt SigLIP for their vision backbones. For instance, PaliGemma (Beyer et al., 2024) integrates it for multimodal generation, while Idefics2 (Laurençon et al., 2024) leverages it for vision-language understanding alongside Llama-based decoders. Cambrian-1 (Tong et al., 2024) employs an ensemble containing SigLIP for vision-centric tasks. Furthermore, NVILA (Liu et al., 2024b) utilizes it for high-resolution image and video processing, as does Qwen3 VL (Bai et al., 2025). Beyond SigLIP, other foundation models like RADIO (Heinrich et al., 2025) have been

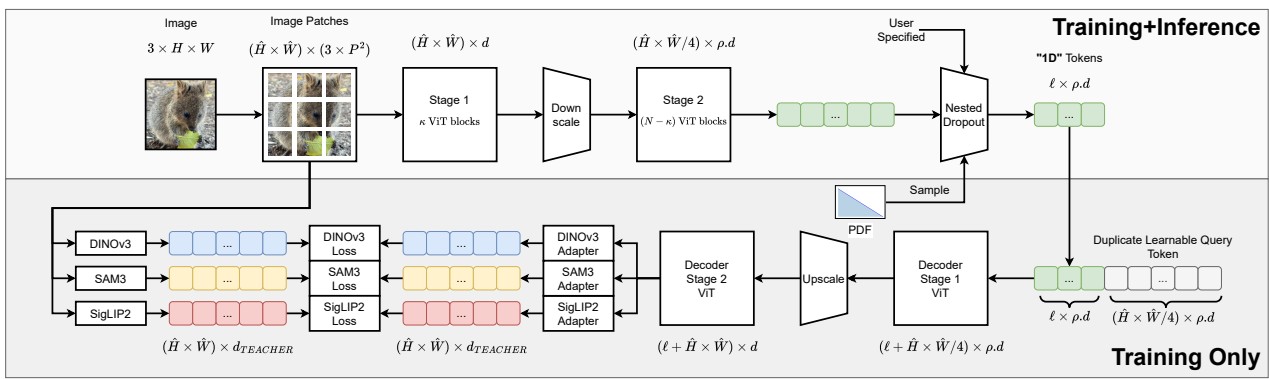

*Figure 2.* Overview of the RADIO1D method. An input RGB image of size $H \times W$ is patchified with patch size $P$ into $\hat{H} \times \hat{W}$ tokens, where $\hat{H} = H/P$ and $\hat{W} = W/P$. Tokens pass through Stage 1, downscaling, Stage 2, and nested dropout to $\ell$ tokens ($\ell$ sampled stochastically during training and specified by the user at inference). During training only, a learnable token is duplicated $\hat{H} \times \hat{W}$ times and concatenated to the $\ell$ tokens, followed by inverse operations: Decoder Stage 1, upscaling, and Decoder Stage 2. The image is also processed by multiple teachers with teacher-specific adapters to compute similarity losses.

adopted by the Nemotron VL (Deshmukh et al., 2025) family to power document intelligence, video understanding, and multi-image reasoning.

The vision encoders CLIP, SigLIP, and SigLIP 2, are predominantly trained using contrastive global objectives for matching entire images with their corresponding captions, facilitating high-level semantic alignment between visual and textual modalities. An exception is C-RADIOv2 (Heinrich et al., 2025), which employs an agglomerative objective by distilling knowledge from multiple foundation models, enabling broader representational capabilities across scales. This raises the question of why popular vision foundation models trained with dense, local objectives such as DINOv3 (Siméoni et al., 2025)'s self-supervised patch-level distillation or SAM3 (Carion et al., 2025)'s promptable segmentation are rarely adopted as backbones in VLMs. A common hypothesis posits image/text contrastive pretraining yields models more suitable for VLMs owing to already being text aligned (Tong et al., 2024; Shi et al., 2025; Wu et al., 2025; Bai et al., 2024; Zhang et al., 2025c), even though the spatial features are not directly supervised with any language objective. If true, it may leave VLMs reliant on encoders with inherently poor dense spatial coherence and explainability, limiting their performance on vision-centric tasks requiring localization or fine-grained reasoning. In this work, we explore this trade-off further by proposing that summarization capabilities rather than raw vision-language alignment may be the most critical factor for effective VLM performance, and we investigate this through novel architectural and training interventions.

Our contributions may be summarized as follows:

- **Analysis of vision features:** We analyze how vision encoders evolve during VLM fine-tuning, showing a shift toward more abstract, less spatially organized representations. We further identify a small set of specialized tokens

in image–text–aligned encoders that capture global image semantics.
- **RADIO1D:** We propose a method that compresses images into flexible, variable-length 1D token sequences using multi-teacher distillation, hierarchical encoding (with early tokens capturing global information), and an autoencoder-like decoder used only during training.
- **Strong summarization:** RADIO1D enables accurate scene understanding with very few tokens—often a single token—demonstrating effective compression of essential visual content.
- **Composition-aware retrieval:** We introduce an evaluation metric that assesses object presence as well as spatial arrangement for retrieved images, and show that compact RADIO1D tokens outperform standard vision encoders.
- **VLM integration:** Integrating RADIO1D into vision-language models yields flexible accuracy–efficiency trade-offs via variable token counts, competitive performance across multimodal tasks, and reduced computational cost.
- **Release:** We release model checkpoints under a permissive license.

The authors are employed by NVIDIA, which leads the development of RADIO.

## 2 Preamble

### 2.1 Analysis of two specialists: SigLIP2 and DINOv3

We analyze two popular vision foundation models: SigLIP2 and DINOv3. SigLIP2 is widely used in VLMs (PaliGemma, Idefics2, Cambrian-1, NVILA, Qwen3-VL) due to strong image-text alignment. DINOv3 excels in dense tasks, achieving state-of-the-art 63.0 mIoU on ADE20K (Zhou et al., 2017) with Mask2Former (Cheng et al., 2022), and extends to detection, depth, and 3D applications (Zhang et al., 2025a; Dou et al., 2026).

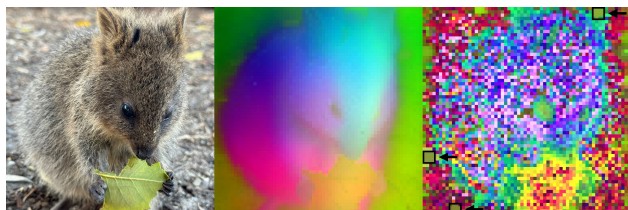

*Figure 3.* PCA visualization of two vision specialists. **Left:** Input Image. **Middle:** DINOv3-H+. **Right:** SigLIP2-SO400m (the three highlighted patches, which appear in the same positions across all images, resemble artifacts at first glance but function as implicit global tokens)

Despite limited crossover—SigLIP2 rarely used for dense tasks, DINOv3 underperforming on OCR—we examine their representations via a PCA projection to 3 channels, then directly map these to RGB (Figure 3). DINOv3 features show clean spatial coherence with color gradients aligned to semantic structures (e.g., fur, leaf). SigLIP2 features are noisier and fragmented, consistent with its global contrastive objective, and suggesting that their masked reconstruction objective doesn't fully recover DINO-like features.

To probe global summarization, we compute per-token ImageNet Top-1 accuracy via k-NN (k=20) on 256×256 images (Figure 4). Center tokens perform better overall (positional bias). DINOv3 outperforms SigLIP2 on average; however, SigLIP2 has three standout tokens (matching PCA anomalies in Figure 3), acting as powerful global summarizers.

We quantify spatial coherence with Centered Kernel Alignment (CKA) (Kornblith et al., 2019) on MS-COCO (Lin et al., 2015) validation embeddings, utilizing the Hilbert-Schmidt Independence Criterion (HSIC):

$$\text{CKA}(\mathbf{X}, \mathbf{Y}) = \frac{\text{HSIC}(\mathbf{K}, \mathbf{L})}{\sqrt{\text{HSIC}(\mathbf{K}, \mathbf{K}) \cdot \text{HSIC}(\mathbf{L}, \mathbf{L})}}$$

where $\mathbf{K} = \mathbf{H}\mathbf{X}\mathbf{X}^\top\mathbf{H}$, $\mathbf{L} = \mathbf{H}\mathbf{Y}\mathbf{Y}^\top\mathbf{H}$, and $\mathbf{H} = \mathbf{I} - \frac{1}{n}\mathbf{1}\mathbf{1}^\top$. Figure 6 shows DINOv3 with bright off-diagonals (strong locality: horizontal/vertical adjacencies). SigLIP2 has weaker correlations and three standout tokens (indices 13, 176, 242). DINOv3 has the highest $\overline{\text{CKA}}_{\text{off}}$; SigLIP2 the lowest. C-RADIOv4 features resemble DINOv3 (strong spatial correlation).

### 2.2 Analysis of a Fine-tuned Vision Encoder

We examine VLM fine-tuning effects (Figure 7). Pre-trained C-RADIOv4 (Ranzinger et al., 2026) features resemble DINOv3 (localized). Post-fine-tuning features become dispersed and noisy, resembling SigLIP2—suggesting VLM optimization discards spatial coherence for cross-modal alignment. Corresponding CKA matrices (Figure 6) show fading off-diagonals post-fine-tuning, with $\overline{\text{CKA}}_{\text{off}}$ dropping from 0.281 to 0.035.

## 3 Method

From an information-theoretic perspective, patch-based image representations impose a fixed 2D grid that is not well matched to the statistics of natural images or the constraints of VLMs. Let an input image $x$ be a realization of a random variable $X \sim p(X)$, with task-relevant information captured by a target variable $Y$. A vision encoder aims to produce a representation $Z$ that maximizes mutual information $I(Z; Y)$ while minimizing representational complexity, commonly quantified by the entropy $H(Z)$. In patch-based encoders, $Z$ consists of spatially localized tokens arranged on a grid. However, natural images exhibit strong spatial correlations, inducing conditional dependencies between neighboring tokens, i.e., $I(Z_i; Z_j \mid X) > 0$. This redundancy increases $H(Z)$, since $H(Z) = \sum_i H(Z_i) - \sum_{i<j} I(Z_i; Z_j) + $ higher-order terms, where the mutual information terms reflect overlapping content across patches. As a result, patch-based representations fail to efficiently compress spatial structure, leading to a suboptimal rate–distortion trade-off under sequence-length and compute constraints.

In contrast, the 1D sequence representation used in RADIO1D enables a more flexible and compressible encoding by treating tokens as an unstructured, elastic sequence. This design aligns with the Information Bottleneck (IB) principle (Tishby et al., 1999), which seeks a representation maximizing $I(Z; Y) - \beta I(Z; X)$. The RADIO1D autoencoder can be viewed as a variational approximation to the IB, mapping $X$ to a variable-length 1D sequence that aggregates global and hierarchical information while reducing redundancy through stochastic slicing and multi-teacher distillation. This enables adaptive compression: visually simpler images require fewer tokens (lower $H(Z)$), while preserving task-relevant information comparable to or exceeding that of rigid 2D grids. Empirically, this manifests as reduced sequence length without degradation in downstream performance, and theoretically corresponds to convergence toward a lower-entropy representation manifold consistent with the minimum description length principle (Rissanen, 1978). In the following paragraphs, we provide more details about our method, which is also depicted in figure 2.

### 3.1 Agglomerative Training

Agglomerative training (Ranzinger et al., 2024) builds on the idea that diverse foundation models can provide complementary representations from large-scale image data, and their knowledge can be distilled into one unified student model. Given an input image $x$, the student's shared backbone generates a summary token $\mathbf{z}^s \in \mathbb{R}^d$ and a set of patch tokens $\mathbf{z}^p_{(i)} \in \mathbb{R}^d$ (for $i = 1, \ldots, N$). For each teacher model $t$ (with its own embedding dimension $d_t$), we attach lightweight adaptor heads—typically small MLPs—to

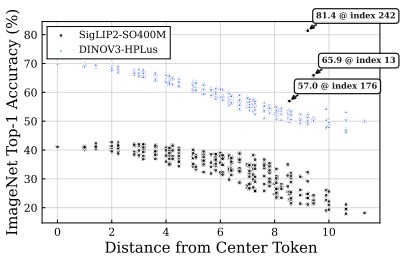 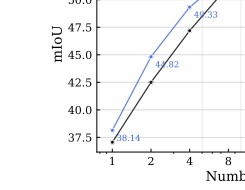 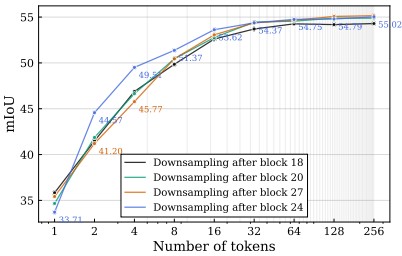

*Figure 4.* Per-token k-NN Top-1 accuracy on ImageNet-1K. SigLIP2 exhibits three strong global classifier tokens.

*Figure 5.* Ablation of the training distribution for $\ell$ (uniform vs. triangular) and downsampling positions $\kappa$. The triangular distribution yields superior ADE20K mIoU overall. We select $\kappa = 24$, as it provides higher ADE20K mIoU on average.

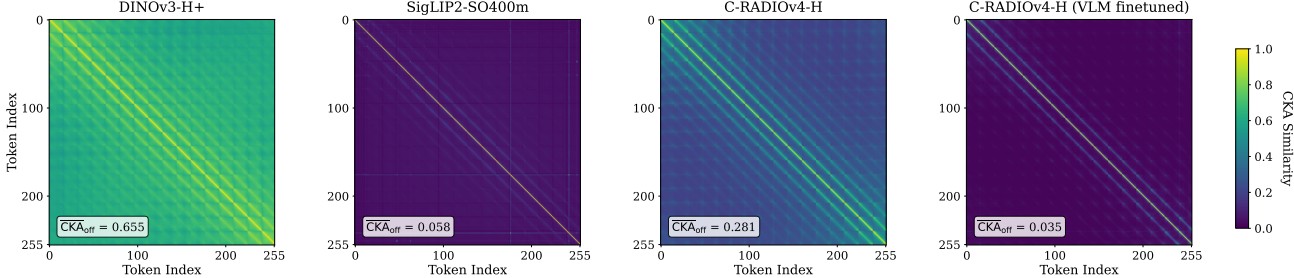

*Figure 6.* CKA matrices, computed on MS-COCO to visualize pairwise token similarities. From left to right: DINOv3-H+, SigLIP2-SO400M, pre-trained C-RADIOv4-H and C-RADIOv4-H (fine-tuned in a VLM). $\overline{\text{CKA}}_{\text{off}}$ denotes the mean off-diagonal value. Pre-trained C-RADIOv4 exhibits a DINOv3-like CKA matrix, but VLM fine-tuning shifts it toward a SigLIP2-like structure.

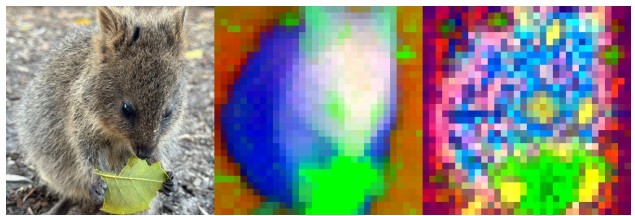

*Figure 7.* PCA visualization of C-RADIOv4 features. **Left:** Input Image. **Middle:** Pre-trained C-RADIOv4-H. **Right:** C-RADIOv4-H after VLM fine-tuning (pre-trained features resemble those of DINOv3, but post-fine-tuning features become dispersed and noisy, resembling SigLIP2).

project the student's features into the teacher's space:

A summary adaptor $g^s_{(t)} : \mathbb{R}^d \to \mathbb{R}^{d_t}$ applied to $\mathbf{z}^s$, yielding $\hat{\mathbf{z}}^s_{(t)} = g^s_{(t)}(\mathbf{z}^s)$. A patch adaptor $g^p_{(t)} : \mathbb{R}^d \to \mathbb{R}^{d_t}$ applied to each $\mathbf{z}^p_{(i)}$, yielding $\hat{\mathbf{z}}^p_{(t,i)} = g^p_{(t)}(\mathbf{z}^p_{(i)})$.

Knowledge distillation then aligns these projected student features with the corresponding summary $\mathbf{z}^s_{(t)}$ and patch features $\mathbf{z}^p_{(t,i)}$ from teacher $t$. The per-teacher loss is

$$\mathcal{L}_t = \ell_s\left(\hat{\mathbf{z}}^s_{(t)}, \mathbf{z}^s_{(t)}\right) + \frac{1}{N}\sum_{i=1}^{N} \ell_p\left(\hat{\mathbf{z}}^p_{(t,i)}, \mathbf{z}^p_{(t,i)}\right),$$

where $\ell_s$ and $\ell_p$ are similarity metrics (e.g., MSE or cosine-based). The total training objective combines losses across all teachers: $\mathcal{L} = \sum_t \lambda_t \mathcal{L}_t$, with weights $\lambda_t$ controlling each teacher's influence. This process enables the student to integrate diverse visual knowledge without modifying its core backbone. We adopt the training recipe of C-RADIOv4

(Ranzinger et al., 2026) to produce RADIO1D.

### 3.2 Student Backbone for 1D Sequence Modeling

Most teacher models process images by dividing them into non-overlapping 2D patches (e.g., 16×16 pixels for patch size 16) and treating the resulting grid as a flattened 1D sequence of patch tokens. This implicit 2D spatial structure poses a challenge when distilling into a student designed under a strict 1D paradigm, where tokens lack predefined spatial arrangement. To bridge this gap while preserving the 1D modeling objective, RADIO1D employs an encoder-decoder architecture within the student backbone. The encoder operates purely in 1D mode: it takes the input image (patched in a standard 2D grid and flattened) and produces a variable-length sequence of 1D tokens with no enforced spatial meaning. These 1D tokens form the core representation that carries all image information forward. The decoder, active only during RADIO training, reconstructs a 2D-compatible grid of patch tokens suitable for alignment with teachers. It is a lightweight Vision Transformer (Dosovitskiy et al., 2021) that receives as input:

- A sequence of duplicated learnable query tokens, whose length matches the number of spatial patches expected by the teachers (e.g., $\frac{H}{P} \times \frac{W}{P}$ for input resolution $H \times W$ and patch size $P$).
- The 1D tokens from the encoder concatenated as additional register tokens.

Cross-attention within the decoder allows the query tokens

to attend to the 1D register tokens, effectively rearranging unstructured 1D information into structured 2D patch features. No direct image or raw patch information bypasses the encoder; the 1D tokens are the sole conduit, ensuring the core model remains 1D-centric.

### 3.3 Elastic 1D Sequence Generation

Inspired by FlexTok (Bachmann et al., 2025), the 1D tokens are obtained by stochastically slicing a variable number of tokens from the encoder's output sequence during training. Specifically, we sample a length $\ell$ from some prior distribution and retain only the first $\ell$ encoder tokens, discarding the rest. This yields an elastic 1D sequence whose length varies across iterations. This mechanism resembles a form of nested dropout: earlier tokens (always preserved) are encouraged to capture global, high-level semantics, while later tokens (subject to higher dropout probability) specialize in finer details. The resulting variability promotes robust hierarchical encoding and prevents overfitting to fixed sequence lengths. Minimizing the objective ensures that $I(T_i : Y) > I(T_j : Y) \quad \forall i < j$. We provide empirical evidence in 4.2, 4.3.

### 3.4 Hierarchical Sequence Downscaling

In many VLMs, vision encoder patch tokens are downscaled via a pixel unshuffle operation, which groups $2 \times 2$ adjacent spatial tokens and concatenates them along the channel dimension, reducing the sequence length by a factor of four before integration with the LLM. This operation is also central to hierarchical vision transformers such as Swin (Liu et al., 2021), where it is referred to as Patch Merging. There, pixel unshuffling expands the embedding dimension from $d$ to $4d$, followed by a linear projection to $\rho d$ channels, with $\rho$ controlling the capacity increase.

Reducing the sequence length from $N$ to $N/4$ yields substantial efficiency gains, particularly at high resolutions (e.g., $1024 \times 1024$ inputs with $N = 4096$ for $16 \times 16$ patches). Transformer complexity consists of a self-attention term $\mathcal{O}(N^2 d)$ and an FFN term $\mathcal{O}(N d^2)$. In regimes where $N \gg d$, the quadratic self-attention term dominates, and downscaling reduces its cost from $\mathcal{O}(N^2 d)$ to $\mathcal{O}(N^2 d/16)$. The FFN cost is reduced from $\mathcal{O}(N d^2)$ to $\mathcal{O}(N d^2/4)$.

In standard Patch Merging configurations (e.g., Swin), the embedding dimension is doubled after unshuffling ($\rho = 2$). Under this setting, self-attention benefits from a net $\sim 8\times$ reduction in compute: the $4\times$ reduction in $N$ yields a $16\times$ speedup, partially offset by a $2\times$ slowdown from increased $d$. In contrast, FFN compute remains approximately unchanged, as the $4\times$ reduction in $N$ is exactly offset by the $4\times$ increase in $d^2$. Overall, Patch Merging substantially reduces per-layer compute, driven primarily by self-attention savings, while enabling higher-capacity representations in later stages.

To internalize this efficiency gain, the RADIO1D encoder integrates Patch Merging blocks directly into its backbone, improving both training and inference efficiency while allowing controlled increases in representational capacity. The decoder, used only during training, applies the corresponding reverse operation (Patch Splitting) to upsample the sequence and restore spatial structure for teacher alignment. We detail the initialization of post-downscaling layers from pre-trained models in Section D.

## 4 Experiments

We fine-tuned the C-RADIOv4-H model using the proposed RADIO1D method, distilling knowledge from teachers SigLIP2-g, DINOv3-7B, and SAM3 with a royalty-free subset of the DataComp1B (Gadre et al., 2023) dataset. Training spanned 300k steps without completing a full epoch, employing a global batch size of 512 low-resolution images (stochastically sampled from resolutions of 128, 192, 224, 256, 384, or 432 pixels) plus 64 high-resolution images (from 512, 768, 1024, or 1152 pixels), resulting in approximately 172M total training samples. This replicates C-RADIOv4's final stage of training, but with half the batch size, to save compute resources.

### 4.1 Ablation Studies

We conduct ablation studies to identify optimal design choices for RADIO1D, focusing on (i) the placement of the downscaling operation, (ii) the embedding expansion factor $\rho$, and (iii) the sampling distribution for the elastic sequence length $\ell$ during training. All ablations are evaluated using ADE20K mIoU as a proxy for semantic segmentation performance. Experiments use a shortened training schedule of 100k iterations (57M samples) and a lightweight SO400M backbone (Alabdulmohsin et al., 2023).

We first perform a static analysis of the design space by comparing parameter count and image throughput across configurations that vary the downscaling position $\kappa$ and expansion factor $\rho$. The downscaling block consists of a $2 \times 2$ pixel unshuffle (expanding patch embeddings to $4d$), layer normalization ($2 \times 4d$ parameters), a linear projection to $\rho d$ channels ($(4d) \times (\rho d)$ parameters), and a projection for the CLS and register tokens ($d \times (\rho d)$ parameters). As shown in Table 5, earlier downscaling improves throughput, while large expansion factors ($\rho > 2$) incur prohibitive parameter and compute costs. We therefore fix $\rho = 2$ for all subsequent experiments.

We next ablate two dynamic hyperparameters: the downscaling position $\kappa$ within the encoder backbone and the training-time sampling distribution for the elastic sequence length $\ell$. For evaluation, we freeze the trained encoders

and apply a linear probe to the reconstructed 2D feature grid produced by the decoder, reporting ADE20K mIoU. Although the decoder is used only for teacher alignment during training, this protocol allows us to assess preservation of spatial-semantic information in a manner consistent with standard encoder evaluations. We vary $\ell$ from 1 to 256 in powers of two.

We first compare sampling distributions for $\ell$ (uniform vs. triangular with Probability Density Function (PDF) $p(x) = 2 - 2x$, see figure 13), fixing $\kappa = 24$. As shown in Figure 5, the triangular distribution more effectively concentrates information into fewer tokens while maintaining strong accuracy at higher token counts. Using this distribution, we then ablate the downscaling position and find $\kappa = 24$ yields the best overall mIoU.

Finally, we evaluate two alternative design choices commonly used in VLMs: (i) removing downscaling entirely, and (ii) applying pixel unshuffling only at the model output, on the 1D token sequence. Both perform substantially worse than configurations with learnable Patch Merging. We attribute this degradation to the reliance of pixel unshuffling on strong spatial locality between neighboring tokens—an assumption that holds for 2D patch grids but breaks down in our 1D tokenization regime, where adjacent tokens exhibit weak or no spatial coherence. Refer to appendix C for more details.

### 4.2 Summarization Capabilities

The RADIO1D model demonstrates strong summarization capabilities by learning to encode a robust global token at index 0, as evidenced by its high per-token k-KNN classification accuracy on ImageNet (figure 12). Unlike figure 4, where the x-axis represents distance to the center, this figure plots accuracy against token index, highlighting that global image information is concentrated in the initial token.

We also show in appendix K that RADIO1D exhibits strong summarization of semantic content even with a **single** token in its bottleneck layer, achieving a high mIoU score of 40.23 on semantic segmentation linear probing using the fully trained version. Notably, the predictions closely align with the ground truth in terms of class identification and spatial positioning, and even finds some elements (cushion, painting) that are missing from the ground truth. We highlight the fact that the mIoU achieved with the **first** token is over four times higher than the mIoU of 8.05 obtained using only the **last** token. This disparity persists across scales: the first 16 tokens yield 54.11 mIoU compared to 20.84 for the last 16. These results underscore the hierarchical ordering inherent in RADIO1D's tokenization, where early tokens capture globally informative features essential for broad semantic understanding, while later tokens encode finer-grained details, enabling efficient performance scaling

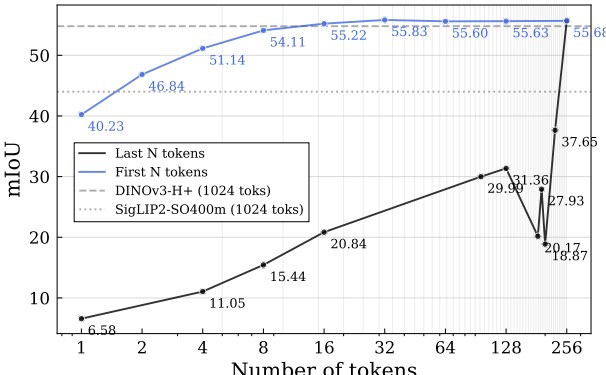

*Figure 8.* ADE20K mIoU as a function of token selection. **Blue:** First N tokens. **Black**: Last N tokens. Nested dropout encourages initial tokens to encode global information and subsequent ones to capture details, yielding smooth additive performance. Selecting last tokens misses global info, with zigzag curve highlighting limitations of opposing the natural training regime.

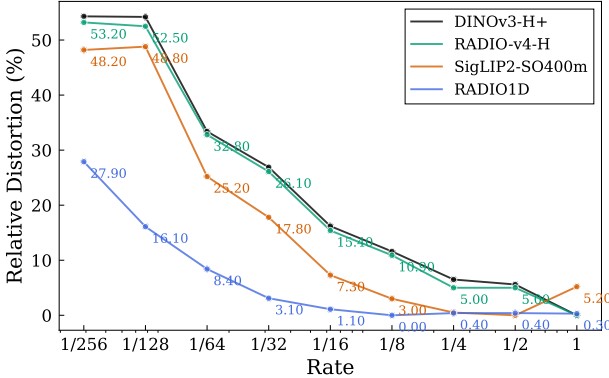

*Figure 9.* Rate-distortion curves on ADE20K semantic segmentation (validation set) using per-model relative distortion $D = (mIoU_{\max} - mIoU)/mIoU_{\max}$, where $mIoU_{\max}$ is each model's performance at full token count. RADIO1D exhibits consistently lower distortion, reflecting the better compressibility of its features.

with minimal computational overhead.

### 4.3 Ties To Information Theory

To empirically validate the information-theoretic motivation underlying RADIO1D, we adopt the classical rate-distortion (R-D) framework from information theory (Cover & Thomas, 2006). In this framework, the rate $R$ corresponds to the complexity of the representation (approximated here by the number of tokens $L$), while the distortion $D$ measures the loss of task-relevant information (quantified as the relative performance degradation $(mIoU_{\max} - mIoU)/mIoU_{\max}$, where $mIoU_{\max}$ is each model's full-resolution performance). This formulation directly aligns with the Information Bottleneck principle (Tishby et al., 1999), which seeks to maximize mutual information $I(Z; Y)$ between the representation $Z$ and the target variable $Y$ (e.g., semantic labels) while minimizing the entropy $H(Z)$. By

plotting R-D curves on ADE20K semantic segmentation using a linear probe decoder, we demonstrate that RADIO1D achieves superior trade-offs–particularly in the low-rate regime–compared to fixed-grid baselines such as SigLIP2, DINOv3, and C-RADIOv4. These baselines are evaluated by subsampling their patch tokens via spatial pooling, allowing a fair comparison of compression efficiency under equivalent representational budgets. As can be seen on figure 9, RADIO1D exhibits consistently lower distortion at all rates. See Appendix E for more details.

## 4.4 Vision-Language Modeling

To assess the utility of RADIO1D in VLMs, we conducted a controlled study within the multimodal Nemotron VL framework (details in appendix H). We isolated the impact of the vision encoder by varying only this component while holding all other hyperparameters, architectural elements, and training configurations constant. Vision encoders were paired with the Nemotron-Nano-9B-v2 (Basant et al., 2025) LLM as the text decoder. Training was performed through the first Supervised Fine-Tuning (SFT) stage on a dataset of 17M multimodal samples drawn from the full Nemotron VL dataset, which comprises diverse image–text pairs curated for multimodal alignment.

We compare RADIO1D at varying token counts against SigLIP2-SO400m, SigLIP2-g, and C-RADIOv4-H baselines. To ensure a fair comparison at an equivalent token budget, the 2D baselines start with 1024 spatial tokens and apply a standard $2 \times 2$ pixel unshuffle to reduce the sequence to exactly 256 tokens before integration with the LLM.

Results are reported in Table 1, demonstrating that RADIO1D outperforms these baselines even with fewer tokens per frame and higher throughput, while offering a continuous spectrum of accuracy–latency trade-offs and greater flexibility in deployment.

Furthermore, evaluating RADIO1D across this elastic spectrum reveals distinct failure modes at extreme compression. Tasks requiring holistic scene understanding such as video reasoning (LongVideoBench) and complex multimodal reasoning (MMMU) are remarkably robust to heavy compression. For instance, MMMU performance drops by less than 10% from its peak score even when the visual representation is reduced to a single token. Conversely, the primary failure mode emerges in dense, reading-heavy tasks. Performance on OCR-centric benchmarks (e.g., DocVQA, InfoVQA, OCRBench) drops sharply as the token budget decreases below 128 tokens, indicating that fine-grained text resolution strictly requires the higher capacity preserved by the full 256-token sequence.

## 4.5 Comparison with Inference-Time Token Pruning

To validate the effectiveness of our learned 1D elastic sequence, we compare RADIO1D against state-of-the-art inference-time token reduction methods. We evaluate two orthogonal approaches applied post-hoc to our dense 2D baseline (C-RADIOv4): Token Merging (ToMe) (Bolya et al., 2023), which combines similar tokens, and CD-Pruner (Zhang et al., 2025b), an MLLM-specific pruning method that utilizes a greedy Determinantal Point Process (DPP) to select a diverse subset of tokens. For CDPruner, we extract vision tokens from the output of the vision projector and text query embeddings from the language model, applying the selection independently on each image tile.

As shown in Table 2, RADIO1D consistently outperforms both ToMe and CDPruner at equivalent token budgets. Notably, at 128 tokens, RADIO1D (71.64% average accuracy) surpasses all CDPruner configurations and even outperforms C-RADIOv4+ToMe operating at 192 tokens (71.49%). We additionally apply CDPruner on top of the full 256-token RADIO1D sequence, but find that natively generating the desired sequence length via our elastic bottleneck is superior.

Crucially, post-hoc pruning and merging methods struggle significantly with dense reading and OCR tasks (e.g., DocVQA, InfoVQA, OCRBench). In failure case analysis for DocVQA, we observed that merging distant tokens degrades spatial awareness; the VLM frequently identifies the correct text element but places it in the wrong layout context (e.g., returning an incorrect cell from a table). These results suggest that learning to compress visual information dynamically during vision backbone training preserves essential spatial grounding far better than training-free token selection or merging.

## 4.6 Composition-Based Retrieval

Noticing that RADIO1D is capable of a 40.23 mIoU on ADE20k with just a single token, it stands to reason that not only are typical semantics of an image encoded, such as "dog", or "chair", but further, the entire scene composition must be roughly encoded in this token. In a typical image retrieval setting, we would look for the embeddings in the database nearest to our query embedding. This is exactly how k-NN is computed. However, we could also test how closely the query and key image scenes are aligned, and thus we present a "composition-based retrieval" metric that aims to identify whether similar objects are present in query and key image, and also whether they're similar in size and location within the image.

Given a query image $q$, we compute an embedding $f(q) \in \mathbb{R}^d$ (e.g. a pooled token, CLS token, or summary token) and retrieve the top-$K$ nearest database images $\{r_1, ..., r_K\}$

| Vision Encoder | Tokens per frame/file | TTFT (ms) | TextVQA (Singh et al., 2019) | DocVQA (Mathew et al., 2020) | InfoVQA (Mathew et al., 2022) | OCRBench (Liu et al., 2024a) | OCRBench2-EN (Fu et al., 2025) | OCRBench2-CN (Fu et al., 2025) | AI2D (Kembhavi et al., 2016) | ChartQA (Masry et al., 2022) | MMMU (Yue et al., 2024) | SeedBench (Li et al., 2024) | LVBench (Wu et al., 2024) | Average |
|---|---|---|---|---|---|---|---|---|---|---|---|---|---|---|
| C-RADIOv4-H | 256 | 468.2 | 84.3 | **93.3** | 77.8 | 84.3 | 60.4 | 41.5 | 86.0 | 89.1 | 50.8 | **78.1** | **58.6** | 73.09 |
| SigLIP2-SO400m | 256 | 440.5 | **85.0** | 92.9 | 77.7 | 82.7 | 61.1 | 40.7 | 86.2 | 88.8 | 50.2 | 77.7 | 56.4 | 72.67 |
| SigLIP2-g | 256 | 517.6 | 84.8 | 92.8 | 77.8 | 82.5 | **62.6** | 41.0 | 86.4 | **89.4** | 49.2 | 77.4 | 57.0 | 72.81 |
| RADIO1D | 1 | 327.0 | 66.2 | 51.1 | 37.8 | 56.5 | 34.1 | 15.2 | 75.6 | 68.2 | 46.9 | 70.0 | 45.2 | 51.53 |
| RADIO1D | 8 | 329.7 | 79.1 | 79.5 | 53.2 | 74.2 | 49.2 | 27.9 | 81.2 | 82.8 | 48.6 | 74.8 | 52.3 | 63.88 |
| RADIO1D | 32 | 335.1 | 81.2 | 88.1 | 62.7 | 77.8 | 55.4 | 34.1 | 84.0 | 85.8 | 48.8 | 76.3 | 55.1 | 68.13 |
| RADIO1D | 64 | 346.1 | 82.0 | 90.6 | 67.7 | 81.2 | 59.5 | 35.2 | 85.0 | 87.9 | 48.4 | 76.9 | 56.3 | 70.07 |
| RADIO1D | 128 | 373.2 | 84.0 | 92.5 | 72.6 | 82.8 | 61.5 | 38.6 | 85.1 | 88.6 | 47.8 | 77.6 | 57.0 | 71.64 |
| RADIO1D | 192 | 380.4 | 84.5 | 93.2 | 75.6 | 83.5 | 60.9 | 40.2 | 85.7 | 89.2 | 48.2 | 77.6 | 57.3 | 72.36 |
| RADIO1D | 224 | 410.2 | 84.4 | 93.0 | 77.3 | **85.0** | 60.2 | 41.8 | 86.4 | 89.2 | **51.4** | 77.8 | 56.8 | 73.02 |
| RADIO1D | 256 | 452.8 | 84.2 | 93.2 | **78.6** | 84.4 | 61.4 | **42.2** | **86.9** | 89.1 | 50.7 | 77.9 | 57.8 | **73.29** |

*Table 1.* Time To First Token and accuracy of vision encoders on 10 multimodal benchmarks when paired with a 9B Nemotron LLM. Results compare C-RADIOv4-H and SigLIP2 baselines (fixed at 256 tokens) against RADIO1D at varying token counts, with average scores reported. TTFT is measured with vLLM on H100 GPUs with 32 images and 128 language tokens in context. See figure 1 for a visualization of the data.

| Vision Encoder | Tokens per tile | TextVQA | DocVQA | InfoVQA | OCRBench | OCRBench2-EN | OCRBench2-CN | AI2D | ChartQA | MMMU | SeedBench | LVBench | Average |
|---|---|---|---|---|---|---|---|---|---|---|---|---|---|
| C-RADIOv4+ToMe | 128 | 82.2 | 90.8 | 69.0 | 81.4 | 58.3 | 36.3 | 84.7 | 87.7 | 49.2 | 76.7 | 57.2 | 70.31 |
| C-RADIOv4+CDPruner | 128 | 82.8 | 89.2 | 69.2 | 74.7 | 52.9 | 33.9 | 83.5 | 87.6 | 50.3 | 77.0 | **58.4** | 69.05 |
| RADIO1D(256)+CDPruner | 128 | 82.2 | 87.2 | 67.6 | 76.5 | 54.6 | 31.9 | 84.2 | 86.8 | **50.4** | 76.2 | 56.3 | 68.56 |
| RADIO1D (ours) | 128 | **84.0** | **92.5** | **72.6** | **82.8** | **61.5** | **38.6** | **85.1** | **88.6** | 47.8 | **77.6** | 57.0 | **71.64** |
| C-RADIOv4+ToMe | 192 | 83.7 | 91.7 | 71.9 | 82.9 | 59.6 | 40.1 | 84.5 | 88.5 | 49.8 | 77.3 | 56.4 | 71.49 |
| C-RADIOv4+CDPruner | 192 | 84.1 | 92.5 | **75.8** | 82.1 | 58.2 | 39.4 | 85.3 | **89.4** | **51.1** | **77.8** | **58.9** | 72.23 |
| RADIO1D(256)+CDPruner | 192 | 83.8 | 92.3 | 75.3 | 82.2 | 59.7 | 39.4 | **86.0** | 88.7 | 50.6 | 77.2 | 57.1 | 72.02 |
| RADIO1D (ours) | 192 | **84.5** | **93.2** | 75.6 | **83.5** | **60.9** | **40.2** | 85.7 | 89.2 | 48.2 | 77.6 | 57.3 | **72.36** |

*Table 2.* Comparison of RADIO1D against post-hoc inference-time token reduction methods (Token Merging and CDPruner) at fixed budgets of 128 and 192 tokens. RADIO1D preserves spatial grounding better for reading-heavy tasks, resulting in higher average accuracy.

using cosine similarity between query and database embedding. Using an object detection dataset, each image is associated with a set of object bounding boxes and categories:

$$\mathcal{B}(q) = \{(c_i, b_i)\}_{i=1}^{n_q}, \quad \mathcal{B}(r) = \{(c'_j, b'_j)\}_{j=1}^{n_r} \quad (1)$$

To quantify composition similarity between $q$ and $r$, we compute the pairwise score matrix $S \in \mathbb{R}^{n_q \times n_r}$ with $R_{ij} = \mathbb{1}[c_i = c'_j]$, $G_{ij} = \frac{\text{gIoU}(b_i, b'_j)+1}{2}$, and $S_{ij} = R_{ij} \odot G_{ij}$, where $\mathbb{1}[\cdot]$ is the indicator function, $\text{gIoU}(x,y) \in [-1,1]$ the generalized IoU (Rezatofighi et al., 2019), and $\odot$ the hadamard product. The matrix $\mathbf{R}$ can be interpreted as the recall matrix, which we also track separately. gIoU is a useful choice because it allows us to encode the difference in object position and size in a single metric. It is preferable compared to $IoU$ because non-overlapping boxes get a score in the range $[-1, 0]$, with increasing values indicating increased alignment. Entries in the score

matrix $\mathbf{S}$ assign no credit when categories mismatch between objects, and partial credit for object match but spatial mismatch, varying smoothly based on the degree of misalignment. We then compute the optimal bipartite assignment using the Hungarian algorithm (Kuhn, 1955): $\mathcal{M}^* = \arg\max_{\mathcal{M}} \sum_{(i,j) \in \mathcal{M}} S_{ij}$, yielding a composition score for the pair $(q, r)$ by normalizing the matched sum:

$$\text{Comp}_{\text{raw}}(q,r) = \sum_{(i,j) \in \mathcal{M}^*} S_{ij} \quad (2)$$

$$r_C^*(q) = \arg\max_{r \in \mathcal{D}} \text{Comp}_{\text{raw}}(q,r) \quad (3)$$

$$\text{Comp}(q,r) = \frac{\text{Comp}_{\text{raw}}(q,r)}{\text{Comp}_{\text{raw}}(q, r_C^*(q))} \quad (4)$$

With $r_C^*(q)$ operating as the oracle result in the database, which maximize the raw score. Then, for each query $q$, we compute the composition score (resp. recall) for the top-$K$ retrieved neighbors and summarize via a max-over-$K$

| Model | MS-COCO | nuImages |
|---|---|---|
| DINOv3 | 0.4813 | 0.3862 |
| SigLIP2 | 0.5051 | 0.4038 |
| C-RADIOv4 | 0.5050 | 0.4046 |
| RADIO1D-H (ours) | **0.5158** | **0.4294** |

*Table 3.* Composition-based retrieval results (Comp@1) on MS-COCO and nuImages. We report the best scoring model configuration from each family: DINOv3-H+, SigLIP2-SO400m-NaFlex, and C-RADIOv4-H. All models worked best when resizing to 512px on the shorter edge, aspect preserving.

operator: $\text{Comp@}K(q) = \max_{1 \leq t \leq K} \text{Comp}(q, r_t)$. Essentially, within the top-$K$ retrieved results, we find the image that maximizes the composition score. Finally, the full benchmark score:

$$\text{Comp@}K = \frac{1}{|Q|} \sum_{q \in Q} \text{Comp@}K(q) \qquad (5)$$

We also calculate $\text{Recall@}K$ with the same formulation, but replace the score matrix $\mathbf{S}$ in (2) with $\mathbf{R}$. We then calculate the composition score for the vision foundation models SigLIP2, DINOv3, C-RADIOv4-H, and finally our proposed RADIO1D. We use MS-COCO and nuImages (Caesar et al., 2019) as evaluation datasets. We show these results in table 3 for $K = 1$, where it can be seen that RADIO1D improves over all other foundation models. We provide more results for each model, as well as recall scores, in section F of the appendix.

## 5 Related Work

Our analysis of specialized tokens builds upon recent observations of token behaviors in Vision Transformers. Darcet et al. (2024) identified high-norm "artifact" tokens that degrade 2D spatial representations and proposed appending "register" tokens to absorb this noise and clean the spatial grid. While we similarly observe outlier tokens, our findings indicate that in image-text aligned models like SigLIP2, these form as low-norm tokens that act as highly effective global summarizers. Consequently, rather than using registers to preserve a 2D grid for dense prediction tasks, RADIO1D leverages this summarization capability to abandon the 2D grid entirely, encoding global and hierarchical information into a compact, elastic 1D sequence.

Recent work on vision foundation models has explored alternatives to fixed 2D patch grids to improve efficiency and flexibility in downstream tasks. Methods such as TiTok (Yu et al., 2024) and Flextok (Bachmann et al., 2025) convert images into compact, variable-length 1D token sequences, enabling more adaptive representations. Like RADIO1D, Flextok employs elastic 1D tokenization and mechanisms such as nested dropout to induce hierarchical ordering, where early tokens capture coarse semantics and later tokens encode finer details. This emphasis on variable-length sequences supports scalable compression while preserving

task-relevant information.

RADIO1D differs in architecture and training objectives. TiTok and Flextok introduce learnable latent or register tokens concatenated with image patches at the transformer input, whereas RADIO1D processes flattened patches through a pure 1D encoder without input augmentation. Moreover, TiTok relies on vector quantization and Flextok on finite scalar quantization, while RADIO1D operates entirely in continuous embedding space to avoid information loss from discretization. Most importantly, TiTok and Flextok are optimized for image reconstruction and generative fidelity (e.g., class-conditional or text-to-image synthesis), whereas RADIO1D uses multi-teacher distillation from diverse foundation models to agglomerate semantically rich features, prioritizing multimodal alignment and vision-centric understanding.

This design enables direct integration of RADIO1D into VLMs, improving efficiency and performance on tasks such as visual question answering and captioning through abstract, spatially decoupled embeddings. In contrast, TiTok and FlexTok do not evaluate their tokenizers within full VLM pipelines for image understanding. RADIO1D therefore bridges dense vision specialists and language-aligned encoders, offering a complementary paradigm for efficient VLM backbones.

## 6 Conclusion

This work challenges the reliance on fixed 2D patch features in vision-language models. Analysis reveals that fine-tuning produces more abstract representations, while image-text aligned encoders like SigLIP2 learn specialized tokens for effective global summarization. We propose RADIO1D, which compresses images into a variable-length 1D token sequence via multi-teacher distillation and autoencoder design. The representations enable strong hierarchical summarization, supporting accurate scene understanding with as few as one token. We introduce a composition-aware image-to-image retrieval benchmark assessing object presence, size, and position, where RADIO1D outperforms strong baselines. In VLMs, it provides flexible accuracy-efficiency tradeoffs via adjustable token counts, competitive multimodal performance, reduced overhead, and improved alignment. Models are released under a permissive license. Future work includes content-aware selection of 1D representation length to adapt token count dynamically to image complexity or task needs, along with extensions to video and further compression techniques. In summary, RADIO1D demonstrates that elastic, summarization-focused representations are the primary alignment that VLMs desire, as opposed to detailed 2D local representations.

## Impact Statement

This paper presents work whose goal is to advance the field of Machine Learning. There are many potential societal consequences of our work, none which we feel must be specifically highlighted here.

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

# A    RADIO1D Architecture Details

In table 4, we show a detailed view of the RADIO1D architecture, with the input shape, parameter count and Floating-Point Operations (FLOPs) associated with each block, assuming an input image size of 512px. RADIO1D blocks match those of the reference RADIO-H model, up to the downsampling block, after which the sequence length is divided by 4, and the embedding dimension multiplied by $\rho$. The net effect is a speed-up and reduction in FLOPs, in favor of the 1D model.

| RADIO1D Encoder (inference) | | | | RADIO1D Decoder (training only) | | | | RADIO (ViT-H/16) (inference) | | | |
|---|---|---|---|---|---|---|---|---|---|---|---|
| Layer | Shape | Params | FLOPs | Layer | Shape | Params | FLOPs | Layer | Shape | Params | FLOPs |
| Input Image | 3×1024×1024 | 0 | 0 | Token Slicing | 1032×2560 | 0 | 0 | Input Image | 3×1024×1024 | 0 | 0 |
| Patchify | 64×64×768 | 0 | 0 | Decoder Tokens (filler + prefix) | 1032×2560 | 2.6M | 0 | Patchify | 64×64×768 | 0 | 0 |
| Patch Projection | 4096×1280 | 984.3K | 4.03G | Decoder Block 0 | 1032×2560 | 78.7M | 167G | Patch Projection | 4096×1280 | 984.3K | 4.03G |
| Add Prefix Tokens (8) | 4104×1280 | 10.2K | 0 | Decoder Block 1 | 1032×2560 | 78.7M | 167.77G | Add Prefix Tokens (8) | 4104×1280 | 10.2K | 0 |
| Add Pos Embed (CPE) | 4104×1280 | 21.0M | 21.0M | Decoder Block 2 | 1032×2560 | 78.7M | 167.77G | Add Pos Embed (CPE) | 4104×1280 | 21.0M | 21.0M |
| Encoder Block 0 | 4104×1280 | 19.7M | 204.49G | PatchSplitting | 1032×2560 | 16.4M | 26.90G | Block 0 | 4104×1280 | 19.7M | 204.49G |
| Encoder Block 1 | 4104×1280 | 19.7M | 204.49G | Decoder Block 3 | 4104×1280 | 19.7M | 204.49G | Block 1 | 4104×1280 | 19.7M | 204.49G |
| Encoder Block 2 | 4104×1280 | 19.7M | 204.49G | Decoder Block 4 | 4104×1280 | 19.7M | 204.49G | Block 2 | 4104×1280 | 19.7M | 204.49G |
| Encoder Block 3 | 4104×1280 | 19.7M | 204.49G | Decoder Block 5 | 4104×1280 | 19.7M | 204.49G | Block 3 | 4104×1280 | 19.7M | 204.49G |
| Encoder Block 4 | 4104×1280 | 19.7M | 204.49G | | | | | Block 4 | 4104×1280 | 19.7M | 204.49G |
| Encoder Block 5 | 4104×1280 | 19.7M | 204.49G | | | | | Block 5 | 4104×1280 | 19.7M | 204.49G |
| Encoder Block 6 | 4104×1280 | 19.7M | 204.49G | | | | | Block 6 | 4104×1280 | 19.7M | 204.49G |
| Encoder Block 7 | 4104×1280 | 19.7M | 204.49G | | | | | Block 7 | 4104×1280 | 19.7M | 204.49G |
| Encoder Block 8 | 4104×1280 | 19.7M | 204.49G | | | | | Block 8 | 4104×1280 | 19.7M | 204.49G |
| Encoder Block 9 | 4104×1280 | 19.7M | 204.49G | | | | | Block 9 | 4104×1280 | 19.7M | 204.49G |
| Encoder Block 10 | 4104×1280 | 19.7M | 204.49G | | | | | Block 10 | 4104×1280 | 19.7M | 204.49G |
| Encoder Block 11 | 4104×1280 | 19.7M | 204.49G | | | | | Block 11 | 4104×1280 | 19.7M | 204.49G |
| Encoder Block 12 | 4104×1280 | 19.7M | 204.49G | | | | | Block 12 | 4104×1280 | 19.7M | 204.49G |
| Encoder Block 13 | 4104×1280 | 19.7M | 204.49G | | | | | Block 13 | 4104×1280 | 19.7M | 204.49G |
| Encoder Block 14 | 4104×1280 | 19.7M | 204.49G | | | | | Block 14 | 4104×1280 | 19.7M | 204.49G |
| Encoder Block 15 | 4104×1280 | 19.7M | 204.49G | | | | | Block 15 | 4104×1280 | 19.7M | 204.49G |
| Encoder Block 16 | 4104×1280 | 19.7M | 204.49G | | | | | Block 16 | 4104×1280 | 19.7M | 204.49G |
| Encoder Block 17 | 4104×1280 | 19.7M | 204.49G | | | | | Block 17 | 4104×1280 | 19.7M | 204.49G |
| Encoder Block 18 | 4104×1280 | 19.7M | 204.49G | | | | | Block 18 | 4104×1280 | 19.7M | 204.49G |
| Encoder Block 19 | 4104×1280 | 19.7M | 204.49G | | | | | Block 19 | 4104×1280 | 19.7M | 204.49G |
| Encoder Block 20 | 4104×1280 | 19.7M | 204.49G | | | | | Block 20 | 4104×1280 | 19.7M | 204.49G |
| Encoder Block 21 | 4104×1280 | 19.7M | 204.49G | | | | | Block 21 | 4104×1280 | 19.7M | 204.49G |
| Encoder Block 22 | 4104×1280 | 19.7M | 204.49G | | | | | Block 22 | 4104×1280 | 19.7M | 204.49G |
| Encoder Block 23 | 4104×1280 | 19.7M | 204.49G | | | | | Block 23 | 4104×1280 | 19.7M | 204.49G |
| PatchMerging | 4104×1280 | 16.4M | 26.90G | | | | | Block 24 | 4104×1280 | 19.7M | 204.49G |
| Encoder Block 24 | 1032×2560 | 78.7M | 167.77G | | | | | Block 25 | 4104×1280 | 19.7M | 204.49G |
| Encoder Block 25 | 1032×2560 | 78.7M | 167.77G | | | | | Block 26 | 4104×1280 | 19.7M | 204.49G |
| Encoder Block 26 | 1032×2560 | 78.7M | 167.77G | | | | | Block 27 | 4104×1280 | 19.7M | 204.49G |
| Encoder Block 27 | 1032×2560 | 78.7M | 167.77G | | | | | Block 28 | 4104×1280 | 19.7M | 204.49G |
| Encoder Block 28 | 1032×2560 | 78.7M | 167.77G | | | | | Block 29 | 4104×1280 | 19.7M | 204.49G |
| Encoder Block 29 | 1032×2560 | 78.7M | 167.77G | | | | | Block 30 | 4104×1280 | 19.7M | 204.49G |
| Encoder Block 30 | 1032×2560 | 78.7M | 167.77G | | | | | Block 31 | 4104×1280 | 19.7M | 204.49G |
| Encoder Block 31 | 1032×2560 | 78.7M | 167.77G | | | | | | | | |
| **Encoder Total** | | **1.14G** | **6280.97G** | **Decoder Total** | | **314.1M** | **1143.69G** | **Total** | | **651.6M** | **6547.84G** |

*Table 4.* Block properties of the RADIO1D encoder and decoder models. The decoder is only used during training. The reference RADIO-H model is also shown for reference.

# B    Static Analysis of the RADIO1D Design Space

In table 5, we show the parameters of the RADIO1D design space affect throughput and parameter counts. We vary the downscaling position $\kappa$ and expansion factor $\rho$. The throughput is measured using vLLM on an H100 GPU with a batch size of 32 and input image size of 1024px. As described in section 3.4, self-attention scales according to $\mathcal{O}(N^2 d)$, and FFN layers as $\mathcal{O}(Nd^2)$. At the downscaling position $\kappa$ the number of tokens changes as $N \to N/4$, and the feature dimension as $d \to \rho d$. When $\rho = 2$, this reduces the self-attention cost by $\sim 8\times$ from $\mathcal{O}(N^2 d) \to \mathcal{O}(2N^2 d/16)$, with no change in the FFN cost from $\mathcal{O}(Nd^2) \to \mathcal{O}(4Nd^2/4)$, which combined leads to fewer flops and higher throughput as $\kappa$ decreases. For $\rho = 4$, the self-attention cost is reduced by $\sim 4\times$ from $\mathcal{O}(N^2 d) \to \mathcal{O}(4N^2 d/16)$, but the FFN cost increases by $\sim 4\times$ from $\mathcal{O}(Nd^2) \to \mathcal{O}(16Nd^2/4)$, which empirically leads to a reduction in throughput as $\kappa$ decreases. Thus from both a parameter and throughput perspective, using $\rho = 2$ is preferred over $\rho = 4$.

# C    Additional Ablations on Downscaling Strategies

To further investigate the importance of learnable Patch Merging integrated early in the RADIO1D encoder backbone, we conducted additional experiments on alternative downscaling strategies commonly employed in VLMs. Before incorporating

| $\kappa$ | Parameter Count | | Throughput (im/s) | |
|---|---|---|---|---|
| | $\rho = 2$ | $\rho = 4$ | $\rho = 2$ | $\rho = 4$ |
| - | 652M | | 51 | |
| 32 | 665M | 652M | 50 | 50 |
| 27 | 963M | 2159M | 55 | 46 |
| 24 | 1140M | 3044M | 58 | 42 |
| 20 | 1376M | 4224M | 63 | 39 |

*Table 5.* Parameter count and throughput (in images/second) of a RADIO1D model, as a function of $\kappa$ and $\rho$.

downscaling directly into the model, we evaluated applying pixel unshuffling operations externally on top of the initial RADIO1D token sequence (which lacked integrated downscaling at the time).

We first attempted the standard 2×2 pixel unshuffling operation (grouping spatially adjacent tokens and concatenating along the channel dimension to reduce sequence length by a factor of 4). The model failed to converge, with training loss remaining persistently high and the model exhibiting severe underfitting. We then explored a flattened one-dimensional variant of unshuffling, for example pivoting every group of 4 consecutive tokens into the channel dimension (4× unshuffling). This approach also failed to produce stable training. Even randomizing the order of the 1D tokens prior to applying the one-dimensional unshuffling marginally improved the training loss but yielded no meaningful convergence gains. These failures highlight that pixel unshuffling critically depends on strong spatial locality among neighboring tokens—a property inherent to 2D patch grids but absent (or very weak) in the 1D tokenization regime of RADIO1D, where adjacent tokens lack coherent spatial relationships.

We additionally compared the initial RADIO1D model (without integrated downscaling) against the final design (with integrated learnable Patch Merging) while fixing the output to 256 1D tokens in both cases. Both variants were evaluated after a short VLM training run on a compact evaluation suite. The model with integrated downscaling achieved superior accuracy across metrics while also offering faster inference (due to reduced sequence lengths in downstream transformer layers, as analyzed in Section 3.4).

| Integrated Downscaling | TextVQA | DocVQA | InfoVQA | OCRBench | VideoMME | LongVideoBench |
|---|---|---|---|---|---|---|
| ✗ | 81.63 | 89.55 | 69.39 | 79.30 | 52.70 | 53.00 |
| ✓ | 82.10 | 91.92 | 73.43 | 80.90 | 54.37 | 56.54 |

*Table 6.* VLM accuracy comparison between initial RADIO1D (no integrated downscaling) and final design (with integrated downscaling), both producing 256 tokens.

These results confirm that learnable Patch Merging integrated into the encoder provides both performance gains and efficiency benefits compared to post-hoc or absent downscaling, consistent with the ablation findings in Section 4.1.

# D    Initialization from Pre-trained Models

We initialize RADIO1D from a pre-trained standard 2D Vision Transformer checkpoint (e.g., C-RADIOv4) using a dedicated conversion procedure. For layers with matching shapes, weights are copied directly. For layers after downscaling (where embedding dimensions or sequence lengths differ), we follow the Net2WiderNet (Chen et al., 2016) approach: we expand source weights by repetition along the expanded dimension, followed by a small amount of random noise to break symmetry. Special handling is applied for stability: output projection weights are divided by the input expansion factor to preserve output magnitude, while LayerNorm parameters are scaled by $\frac{1}{\sqrt{\rho}}$ to maintain variance.

# E    Rate-Distortion Analysis

## E.1    Rate-Distortion Framework for Lossy Compression Analysis

We evaluate RADIO1D's information-theoretic motivation using the classical rate-distortion (R-D) framework (Shannon, 1959; Cover & Thomas, 2006). Here, the rate $R$ approximates the complexity or entropy $H(Z)$ of the learned representation $Z$ (proxied by token count $L$). The distortion $D$ quantifies loss of task-relevant information, defined as the normalized performance degradation $D = \frac{mIoU_{\max} - mIoU}{mIoU_{\max}}$, where $mIoU_{\max}$ is the model's performance at full token count ($L = 256$ or equivalent). This formulation directly aligns with the Information Bottleneck principle (Tishby et al., 1999), which seeks to maximize mutual information $I(Z;Y)$ between representation $Z$ and target variable $Y$ (semantic labels) while minimizing $H(Z)$. RADIO1D's nested dropout and hierarchical design are intended to produce a superior R-D frontier by concentrating task-relevant information in early tokens and discarding spatial redundancy.

## E.2    Experimental Setup

We use the ADE20K dataset and a frozen linear-probe decoder. For RADIO1D, we vary the slice length $L \in \{1, 4, 16, 64, 256\}$. For fixed-grid baselines (DINOv3-H+, SigLIP2-SO400m, C-RADIOv4-H), we reduce the original 32×32 (1024-token) grid to the target effective token count via spatial average pooling (preserving 2D locality). We show these results in table 7. RADIO1D consistently exhibits the lowest relative distortion across all rates, with a particularly pronounced advantage in the low-rate regime (e.g., at $L = 1$, distortion 27.9% vs. 72-76.7% for baselines). This supports the efficacy of hierarchical encoding: early tokens capture most semantic information, enabling high accuracy with minimal representational budget.

| Rate | DINOv3-H+ | | | C-RADIOv4-H | | | SigLIP2-SO400m | | | RADIO1D | | |
|---|---|---|---|---|---|---|---|---|---|---|---|---|
| | Tokens | mIoU | Distortion | Tokens | mIoU | Distortion | Tokens | mIoU | Distortion | Tokens | mIoU | Distortion |
| 1 | 1024 | 54.80 | **0.0%** | 1024 | 55.20 | **0.0%** | 1024 | 44.00 | 5.2% | 256 | **55.68** | **0.3%** |
| 1/2 | 512 | 51.73 | 5.6% | 512 | 52.45 | 5.0% | 512 | 46.39 | **0.0%** | 128 | **55.63** | **0.4%** |
| 1/4 | 256 | 51.24 | 6.5% | 256 | 52.43 | 5.0% | 256 | 46.18 | **0.5%** | 64 | **55.60** | **0.4%** |
| 1/8 | 128 | 48.42 | 11.6% | 128 | 49.16 | 10.9% | 128 | 44.99 | 3.0% | 32 | **55.83** | **0.0%** |
| 1/16 | 64 | 45.94 | 16.2% | 64 | 46.69 | 15.4% | 64 | 43.02 | 7.3% | 16 | **55.22** | **1.1%** |
| 1/32 | 32 | 40.06 | 26.9% | 32 | 40.79 | 26.1% | 32 | 38.13 | 17.8% | 8 | **54.11** | **3.1%** |
| 1/64 | 16 | 36.52 | 33.4% | 16 | 37.08 | 32.8% | 16 | 34.71 | 25.2% | 4 | **51.14** | **8.4%** |
| 1/128 | 8 | 25.11 | 54.2% | 8 | 26.23 | 52.5% | 8 | 23.76 | 48.8% | 2 | **46.84** | **16.1%** |
| 1/256 | 4 | 25.07 | 54.3% | 4 | 25.83 | 53.2% | 4 | 24.04 | 48.2% | 1 | **40.23** | **27.9%** |
| 1/512 | 2 | 12.43 | 77.3% | 2 | **13.17** | 76.1% | 2 | 12.78 | **72.5%** | – | – | – |
| 1/1024 | 1 | 12.76 | 76.7% | 1 | **13.13** | 76.2% | 1 | 12.88 | **72.2%** | – | – | – |

*Table 7.* Rate-Distortion analysis, applied to DINOv3-H+, C-RADIOv4-H and SigLIP2-SO400m.

# F    Composition-Based Retrieval (Extended)

For the models that support it, we tried both square center crops, as well as aspect preserving resizing modes, and found, also intuitively, that aspect preserving processing improves the results for all models except for DINOv3-7B on MS-COCO. We can also see that the composition score isn't strictly dominated by improved recall scores, as SigLIP2-SO400M-NaFlex often achieves the best recall scores, while having lower composition scores than RADIO1D. Also, owing to the fact that RADIO1D produces an ordered set of tokens, we study whether using the first two tokens, by concatenating them in the channel dimension, works better than using solely the first, the "512min_T2" setting. We observe that there is a slightly positive effect for MS-COCO, but negative effect on nuImages, suggesting that this technique is unlikely to be beneficial. We show these results in table 8. We also show qualitative results in figure 10 for MS-COCO, and figure 11 for nuImages. nuImages contains a large degree more examples of roughly the same scene, with variations in objects present.

Query Image

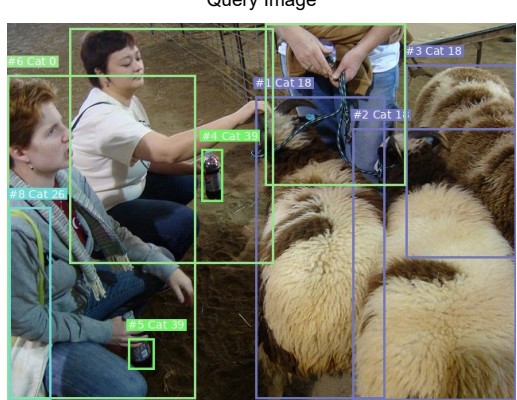

Best Match in Database (Oracle)

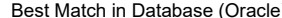
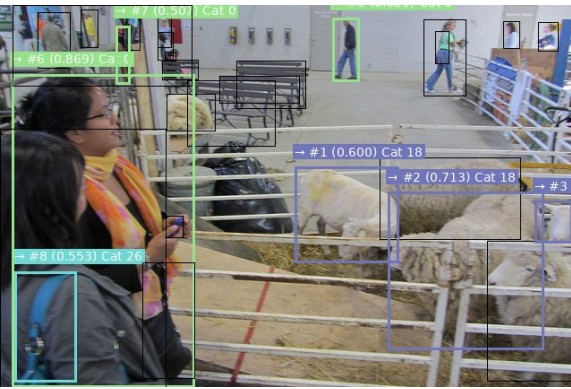

RADIO1D (Score: 1.0)

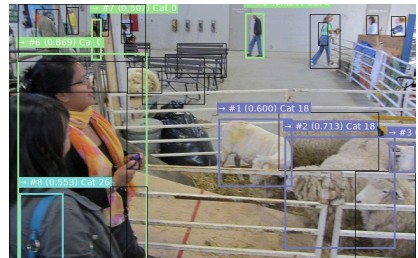

C-RADIOv4-H (Score: 0.74)

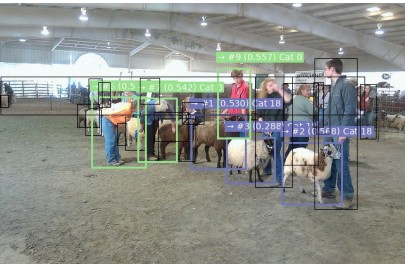

SigLIP2-g-384 / SO400M (Score: 0.74)

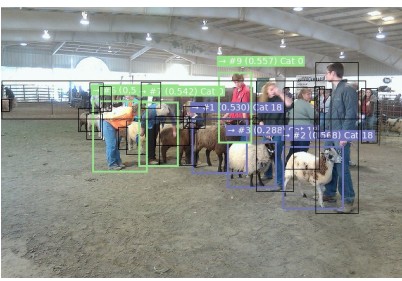

DINOv3-H+ (Score: 0.45)

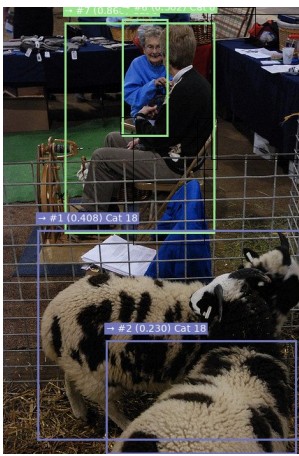

DINOv3-7B (Score: 0.32)

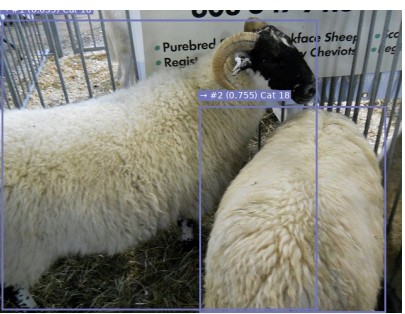

*Figure 10.* Qualitative composition retrieval results for MS-COCO. Query images come from the val set, and database is the training set. For the oracle and model retrieved results, for each object, we show which query object got assigned "x", the gIoU score, and the category label. RADIO1D exactly retrieves the oracle image.

Query Image

Best Match in Database (Oracle)

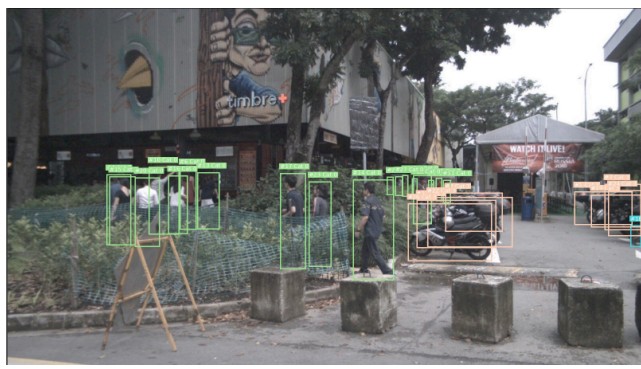
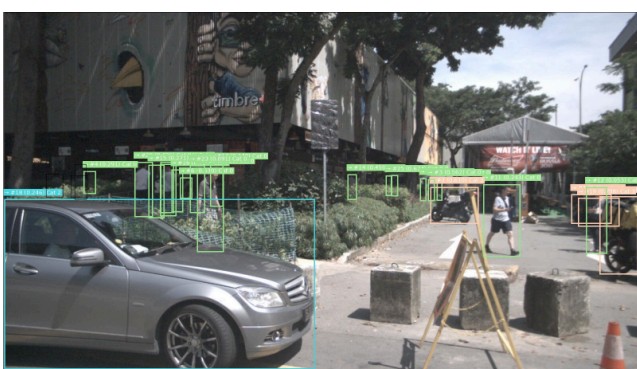

RADIO1D (Score: 0.76)

C-RADIOv4-H (Score: 0.74)

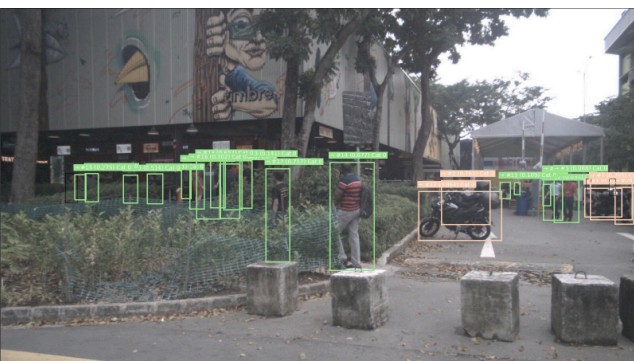
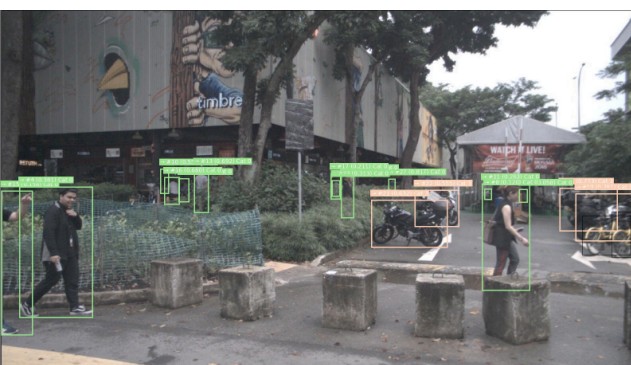

SigLIP2-g-384 / SO400M (Score: 0.65)

DINOv3-7B / H+ (Score: 0.46)

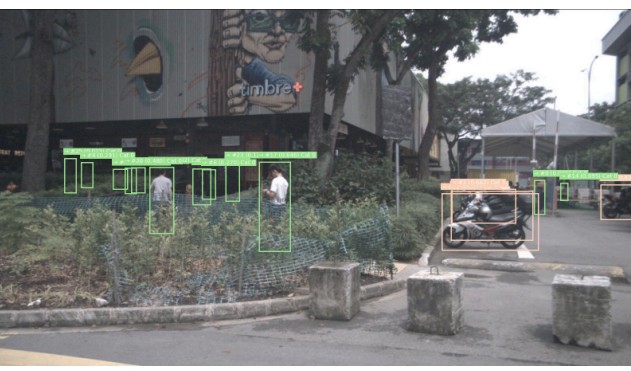
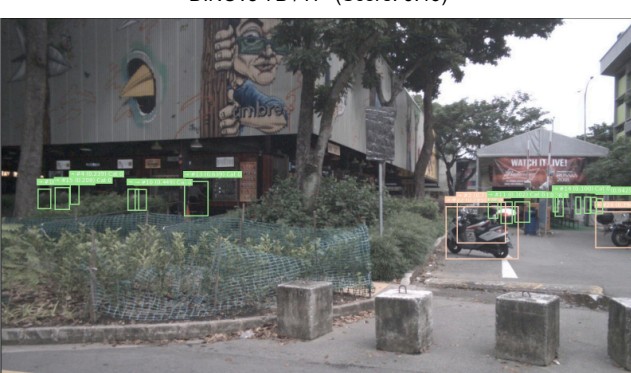

*Figure 11.* Qualitative composition retrieval results for nuImages. Query images come from the val set, and database is the training set. For the oracle and model retrieved results, for each object, we show which query object got assigned "x", the gIoU score, and the category label.

| Model | Resolution | MS-COCO | | | | nuImages | | | |
|---|---|---|---|---|---|---|---|---|---|
| | | Comp@$K$ | | Recall@$K$ | | Comp@$K$ | | Recall@$K$ | |
| | | Top-1 | Top-5 | Top-1 | Top-5 | Top-1 | Top-5 | Top-1 | Top-5 |
| DINOv3-7B | 512x512 | 0.4795 | 0.6391 | 0.6435 | 0.8009 | 0.3844 | 0.5470 | 0.6251 | 0.8452 |
| DINOv3-7B | 512min | 0.4781 | 0.6403 | 0.6434 | 0.8008 | 0.3858 | 0.5488 | 0.6315 | 0.8497 |
| DINOv3-H+ | 512x512 | 0.4812 | 0.6427 | 0.6474 | 0.8047 | 0.3834 | 0.5484 | 0.6248 | 0.8480 |
| DINOv3-H+ | 512min | 0.4813 | 0.6423 | 0.6483 | 0.8086 | 0.3862 | 0.5512 | 0.6334 | 0.8507 |
| SigLIP2-G | 384x384 | 0.4981 | 0.6622 | 0.6821 | 0.8342 | 0.3939 | 0.5586 | 0.6442 | 0.8617 |
| SigLIP2-SO400M-512 | 512x512 | 0.4988 | 0.6619 | 0.6812 | 0.8344 | 0.3990 | 0.5608 | 0.6433 | 0.8598 |
| SigLIP2-SO400M-NaFlex | 512x512 | 0.4971 | 0.6585 | 0.6796 | 0.8338 | 0.3966 | 0.5596 | 0.6464 | 0.8624 |
| SigLIP2-SO400M-NaFlex | 512min | 0.5051 | 0.6641 | **0.6936** | 0.8395 | 0.4038 | 0.5683 | **0.6692** | **0.8752** |
| C-RADIOv4-H | 512x512 | 0.5023 | 0.6637 | 0.6812 | 0.8325 | 0.3986 | 0.5610 | 0.6412 | 0.8561 |
| C-RADIOv4-H | 512min | 0.5050 | 0.6640 | 0.6909 | 0.8372 | 0.4046 | 0.5638 | 0.6597 | 0.8648 |
| RADIO1D-H (ours) | 512x512 | 0.5135 | 0.6762 | 0.6795 | 0.8360 | 0.4243 | 0.5829 | 0.6441 | 0.8567 |
| RADIO1D-H (ours) | 512min | 0.5158 | 0.6787 | 0.6933 | **0.8440** | **0.4294** | **0.5884** | 0.6555 | 0.8663 |
| RADIO1D-H (ours) | 512min_T2 | **0.5164** | **0.6803** | 0.6869 | 0.8377 | 0.4234 | 0.5829 | 0.6478 | 0.8612 |

*Table 8.* Composition-aware image-to-image retrieval results on MS-COCO (Lin et al., 2015) and nuImages (Caesar et al., 2019). We report gIoU-based composition scores and recall at Top-$K$ (Top-1/Top-5). For $X$ min resolution, we resize the minimum-length edge to $X$ while preserving aspect ratio and rounding the long edge to the nearest size divisible by the model patch size. For RADIO1D variants with "_T#", we concatenate the first # encoder tokens into a single vector for retrieval.

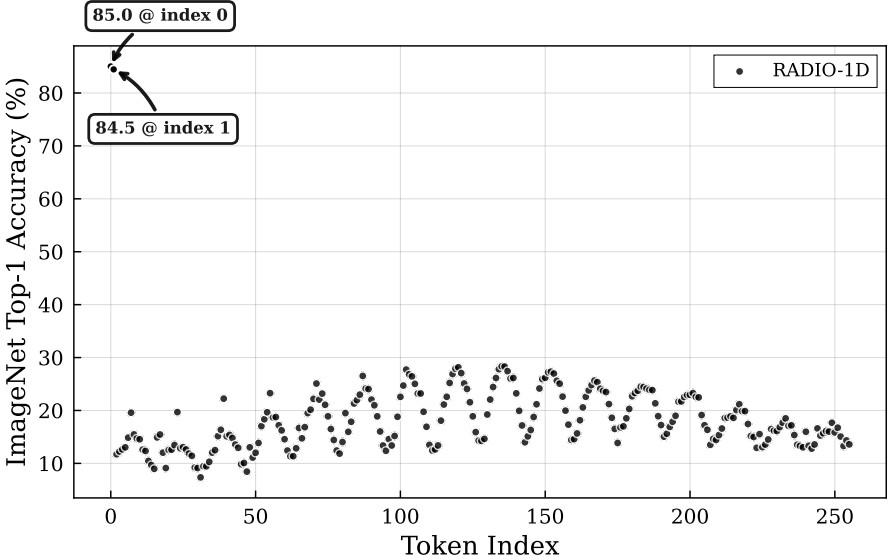

*Figure 12.* Per-Token KNN Top-1 Accuracy on ImageNet1k.

# G   Per-Token k-NN ImageNet-1k Top-1 Accuracy

In the above figure, we show the summarization capabilities of the first RADIO1D tokens: the first two tokens achieve a k-NN Top-1 ImageNet-1k classification accuracy of 85.0% and 84.5% respectively. We note that the following tokens still exhibit mild spatial correlations: tokens that map to near-center locations of the image show slightly better accuracy, and the oscillations are still visible with a period of 16 tokens, corresponding to the width of the 512px input image.

## H    Nemotron VL Framework

We adopted key elements of the training protocol from Nemotron VL 2 (Deshmukh et al., 2025), leveraging the Megatron [1] framework for distributed training, Transformer Engine [2] for optimized mixed-precision computations, and the Megatron Energon [3] dataloader for efficient data loading and preprocessing.

Similar to InternVL (Chen et al., 2023), we employ image tiling to support variable input resolutions while approximately preserving aspect ratio. We use a tile size of 512px with a maximum of 12 tiles, in addition to a "thumbnail" tile obtained by resizing the original image to a square of 512px. For SigLIP2-G-384, originally trained at 384px resolution, we apply positional encoding interpolation to 512px.

For 2D vision encoders, we apply a $2 \times 2$ pixel unshuffle to reduce the number of vision tokens by a factor of 4. Since RADIO1D already integrates a similar downscaling operation internally, we skip the additional pixel unshuffle for this vision encoder.

Training proceeds in two stages. In the first pre-training stage (Nemotron VL Stage 0), we train only the vision-to-language projector while keeping the vision encoder and language model frozen. In the second stage, we unfreeze all components and perform supervised fine-tuning (SFT) on a dataset of 17M samples (image–text subset of the full 25M Nemotron VL dataset, which additionally includes text-only and video data).

We train with a maximum sequence length of 16,384 tokens and apply data packing during SFT to improve efficiency, achieving ∼5 samples per batch per data-parallel rank. For vision encoders that produce fewer output tokens (e.g., RADIO1D with 32 tokens instead of the full 256), we reduce the packing size to maintain ∼5 samples per batch. This enables a fair comparison across vision encoders with varying token counts without modifying hyperparameters such as the learning rate or training schedule.

We omit Nemotron VL's Stage 2 (49k context extension), Stage 3 (49k text recovery), and Stage 4 (300k context extension).

## I    Distribution Visualizations

In figure 13 we plot the PDFs of the uniform and triangular distribution $(2 - 2x)$ studied in the ablations. We also plot the survival function $f(y) = P(X \geq y)$, which is:

$$f_{uni}(y) = 1 - y \tag{6}$$

$$f_{tri}(y) = 1 - 2y + y^2 \tag{7}$$

for the uniform and our triangular distribution respectively. The triangular distribution allocates much more probability density toward lower token counts, clearly having an effect on semantic segmentation at lower token counts, as seen in figure 5. Interestingly, the likelihood of high token counts is much lower for the triangular distribution, likely explaining why mIoU stops improving after 32 tokens, however, it still is able to match the quality of the uniform distribution model at these higher counts.

## J    SigLIP2 Feature Visualizations

In figure 14 we present visualizations of the SigLIP2-SO400m model's token embeddings for four diverse input images, each resized to 512×512 pixels, resulting in a 32×32 grid of tokens. The middle column displays heatmaps of the token norms, where darker regions indicate lower-norm values. Notably, across all images (regardless of content) three consistent groups of tokens exhibit markedly lower norms: one positioned near the top, slightly inset from the right border; another along the left border, approximately two-thirds down vertically; and a third at the bottom, roughly one-quarter of the way from the left horizontally. These same token groups also stand out prominently in the right column's PCA projections of the embeddings into RGB space, appearing as distinct outliers in color and pattern compared to the surrounding tokens. As discussed in the main text, these invariant positions correspond to the "global" tokens we identified, which appear to capture global image information rather than localized visual features.

---

[1] https://github.com/NVIDIA/Megatron-LM
[2] https://github.com/NVIDIA/TransformerEngine
[3] https://github.com/NVIDIA/Megatron-Energon

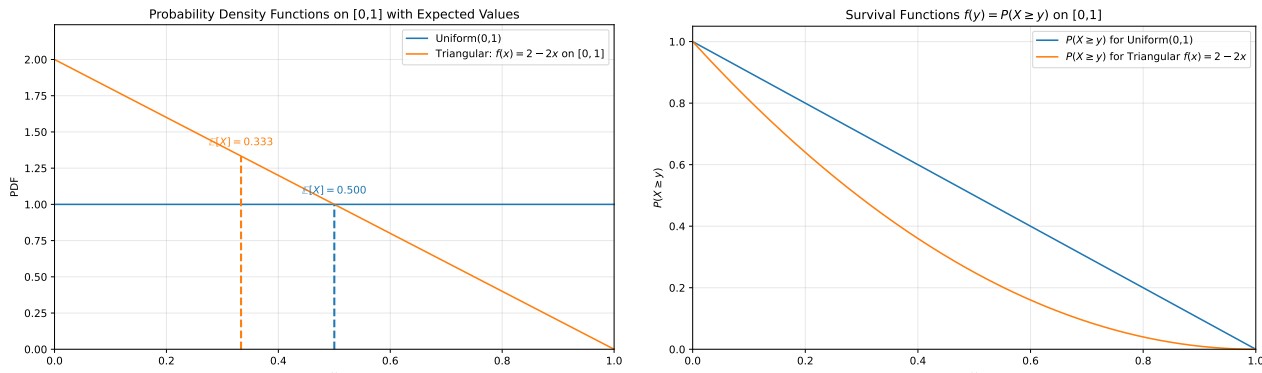

*Figure 13.* **Left:** The PDF for the uniform distribution and $2 - 2x$ triangular distribution. **Right:** The survival probability for a given token index for the respective distributions.

## K   Semantic Segmentation Visualizations

The visualizations in figure 15 demonstrate the effectiveness of our semantic segmentation model on ADE20K validation scenes, comparing ground truth annotations (left) against predictions using a single token per 512×512 crop in the bottleneck (middle) and 256 tokens per crop (right). Remarkably, even with just one token, the model reconstitutes a substantial portion of the scene's semantic content, accurately identifying major elements such as cars, buildings, trees, and furniture, though with vague, blob-like outlines that approximate shapes without fine boundaries. For instance, in the urban street scene, parked cars appear as broad blue masses merging into the road, while in the indoor living room, armchairs and tables form rough cyan and pink clusters. This resembles a semantic segmentation of low-resolution version of the input image, despite the model never being trained to explicitly capture such information. Scaling to 256 tokens significantly enhances fidelity, yielding highly detailed shapes that closely mirror the ground truth; standout improvements include precise delineations of slender objects like streetlights and flags in outdoor views, intricate furniture contours such as individual cushions on armchairs or curved lamp shades in the living room, and subtle bedroom elements like pillows, fans, and windowpanes, enabling near-photorealistic semantic reconstructions that capture nuanced spatial relationships and object intricacies.

## L   CKA Regularization

Following from the analysis in section 2.1, with DINOv3 having a much higher off-diagonal CKA versus SigLIP2 (figure 6), along with each patch generally operating well as a classifier (figure 4), it implies that the representations of DINOv3 are highly redundant. For dense spatial tasks, especially with a frozen backbone, this is a feature, not a bug, because it means that each token is encoding information that is local and spatially coherent. Given the bottleneck structure of RADIO1D, it gives us the flexibility to encourage the encoder representations to behave a certain way directly. For this, we formulate the off-diagonal mean CKA as a regularization term directly:

$$\mathrm{CKA}_\ell(\mathbf{X}, \mathbf{Y}) = \frac{\|\mathbf{X}^\top \mathbf{Y}\|_F^2}{\|\mathbf{X}^\top \mathbf{X}\|_F^2 \|\mathbf{Y}^\top \mathbf{Y}\|_F^2} \tag{8}$$

$$\mathrm{CKA}_{\mathrm{reg}}(\boldsymbol{\mathcal{X}}) = \frac{\gamma}{\ell(\ell-1)} \sum_{\substack{i,j \in \ell \\ i \neq j}} \mathrm{CKA}_\ell(\boldsymbol{\mathcal{X}}_i, \boldsymbol{\mathcal{X}}_j) \tag{9}$$

with $\mathrm{CKA}_\ell$ being the linear CKA kernel, and $\mathrm{CKA}_{\mathrm{reg}}$ being the regularization formula, with $\gamma$ being the loss weight. Given that $\mathbf{X}, \mathbf{Y} \in \mathbb{R}^{B \times D}$ are matrices in (8), then $\boldsymbol{\mathcal{X}} \in \mathbb{R}^{B \times \ell \times D}$ is the output of the encoder, with batch size $B$, tokens $\ell$, and dimension $D$. So, we construct the $\ell \times \ell$ cartesian product of CKAs between position $i$ and $j$ s.t. $i \neq j$, and then average. Given that $B = 576$, $\ell \in \{1, 2, ..., 1296\}$, and $D = 1152, 1280$ for SO400M and H respectively, this becomes a very computationally heavy operation: $O(B^2\ell^2 D^2)$. To make this something can be tractably computed online during training, we stochastically reduce the size of $\boldsymbol{\mathcal{X}}$ in the following way:

- We reduce $D \to \hat{D}$ with a random $D \times \hat{D}$ projection matrix, relying on the JL lemma for approximate distance preservation.

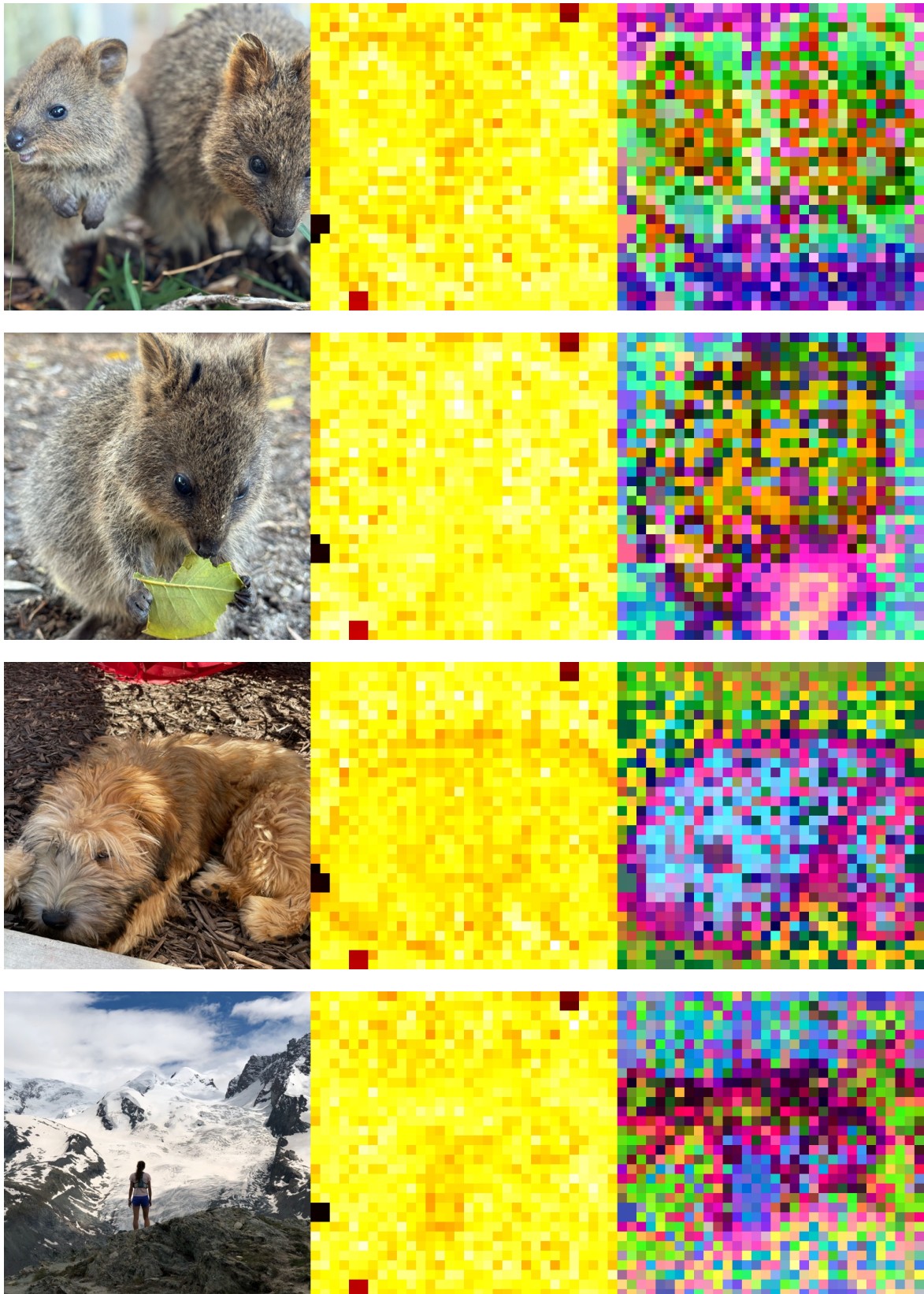

*Figure 14.* Visualization of SigLIP2-SO400m features. **Left:** Input image. **Middle:** Heatmap of the token norms. **Right:** PCA projection of token embeddings to RGB channels.

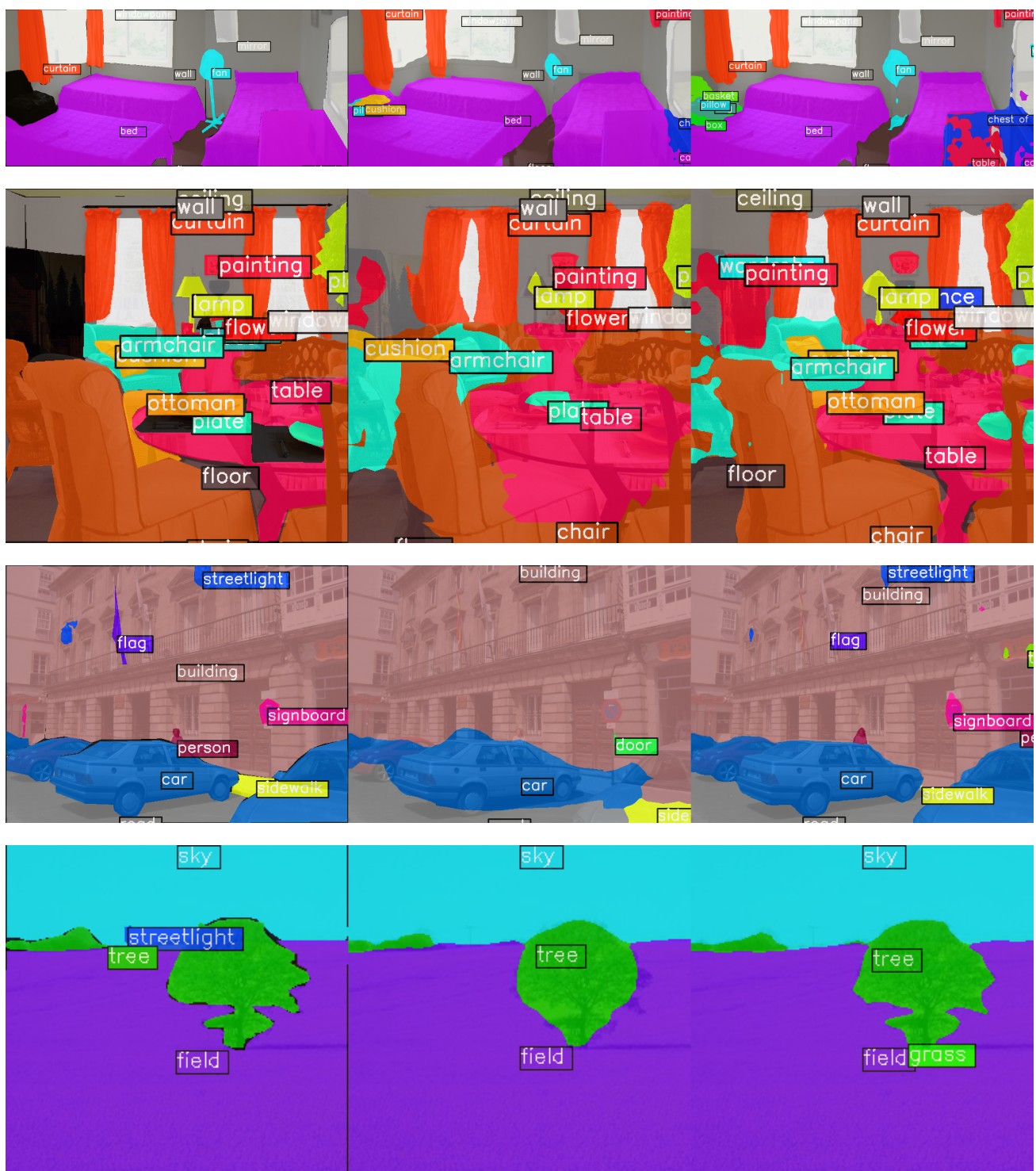

*Figure 15.* Visualization of ADE20k ground truth and predictions. **Left**: Ground Truth. **Middle**: Prediction with 1 token per 512 × 512 crop in the bottleneck. **Right**: Prediction with 256 tokens per 512 × 512 in the bottleneck.

- We reduce $B \to \hat{B}$ by randomly sampling a subset from $B$.

- We reduce $\ell \to \hat{\ell}$ by randomly sampling a subset from $\ell$.

We provide the pseudocode in section L.1. In figure 16 we show the CKA visuals for the baseline 2D model, as well as without, and with, the CKA regularization. It is immediately clear that the regularizer works as intended, as the off-diagonal relationships are nearly 0. It's also apparent that without CKA, the model retains a lot of 2D structure, albeit with much less energy. It appears to be the case that a lot of the correlations in 2D are between consecutive rows in the prefix, and much lower correlation between consecutive tokens horizontally.

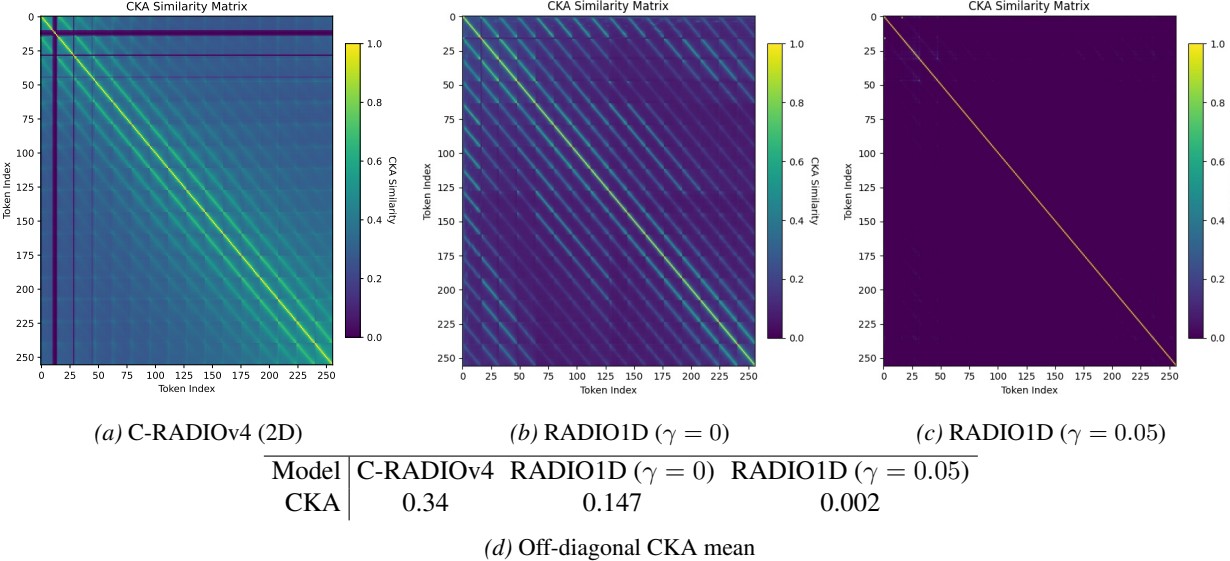

| | | | |
|---|---|---|---|
| *(a)* C-RADIOv4 (2D) | *(b)* RADIO1D ($\gamma = 0$) | *(c)* RADIO1D ($\gamma = 0.05$) |

| Model | C-RADIOv4 | RADIO1D ($\gamma = 0$) | RADIO1D ($\gamma = 0.05$) |
|---|---|---|---|
| CKA | 0.34 | 0.147 | 0.002 |

*(d)* Off-diagonal CKA mean

*Figure 16.* **(A-C)** We show the CKA plots for the baseline C-RADIOv4 model, which is a 2D model, followed by RADIO1D without CKA regularization, and finally with CKA regularization. **(D)** We report the mean off-diagonal CKA.

To see if this structure is important, with a new training run we continually adjusted whether the CKA was applied at various points in training, starting off with $\gamma = 0.05$ to "break" the 2D structure, and then decay to $\gamma = 0$, allowing unconstrained correlations to emerge. In figure 19 we show the CKA plots for this model. Notably, these 2D structures are actually quite sticky, as [19b] and [19e] are nearly the same. Because we were able to reduce it to nearly 0 before epoch 30, relaxed it by 30, constrained again by epoch 50, and fully relaxed again by epoch 140, the learning process appears to specifically prefer this structure.

In figure 17 we plot the ADE20k mIoU as a function of the number of tokens preserved in the encoder. We can see that while the representations must have very different structure, it has an insignificant effect on the decoder's ability to reconstruct a semantic image. Notably, the model without CKA achieves an mIoU of 55.66 with just 32 tokens, nearly matching DINOv3-7B at $< \frac{1}{10}$ the parameters, and surpassing the baseline C-RADIOv4-SO400m 2D model, which achieved 55.14 using 1024 tokens. This appears to lend credence to the conjecture in (Yu et al., 2024) that an image is worth 32 tokens.

In figure 18 we study two different reconstruction modes, where we measure the MSE between the decoder given some subset of tokens, versus the full number of tokens. In [18a] we use a prefix of length K, where we'd expect the reconstruction error to decrease as the number of tokens increases. We can see that initially the "CKA 0.05" model does marginally better at self-reconstruction, up to token 16, and then does not improve on the error at nearly the same rate as the "CKA 0" model. In [18b] we instead use only the token at index M to perform the reconstruction. Unsurprisingly, by construction, M=0 has the lowest error. For $M > 0$, we are no longer dealing with a prefix, and thus the most informative tokens are missing. Instead, what we see is that the "CKA 0.05" model has a consistently lower reconstruction error for any given token. While the CKA loss term was successful at strongly reducing these correlations, it failed to lead to any measurable improvement on downstream metrics (e.g. figure 17), and curiously leading to potentially increased redundancy, seen in figure 18b, and so we don't include it during regular model training.

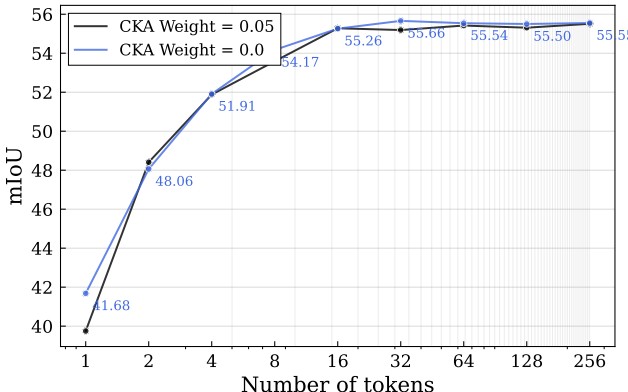

*Figure 17.* ADE20k linear probe. We show two models, trained with or without CKA regularization, and the mIoU given a certain number of prefix tokens.

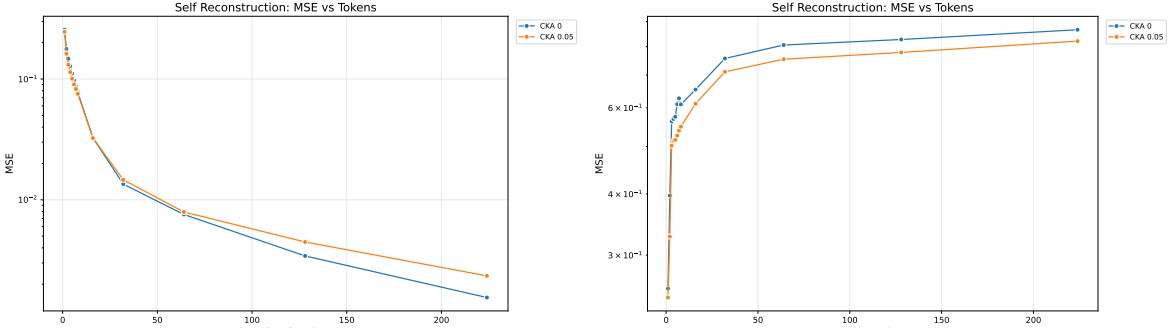

*(a)* The MSE between the decoder given a prefix of K tokens, versus itself given the full 256 tokens.

*(b)* The MSE between the decoder given a particular token at index M, versus itself given the full 256 tokens.

*Figure 18.* Two reconstruction plots, measuring the ability for the decoder to reconstruct its own representations.

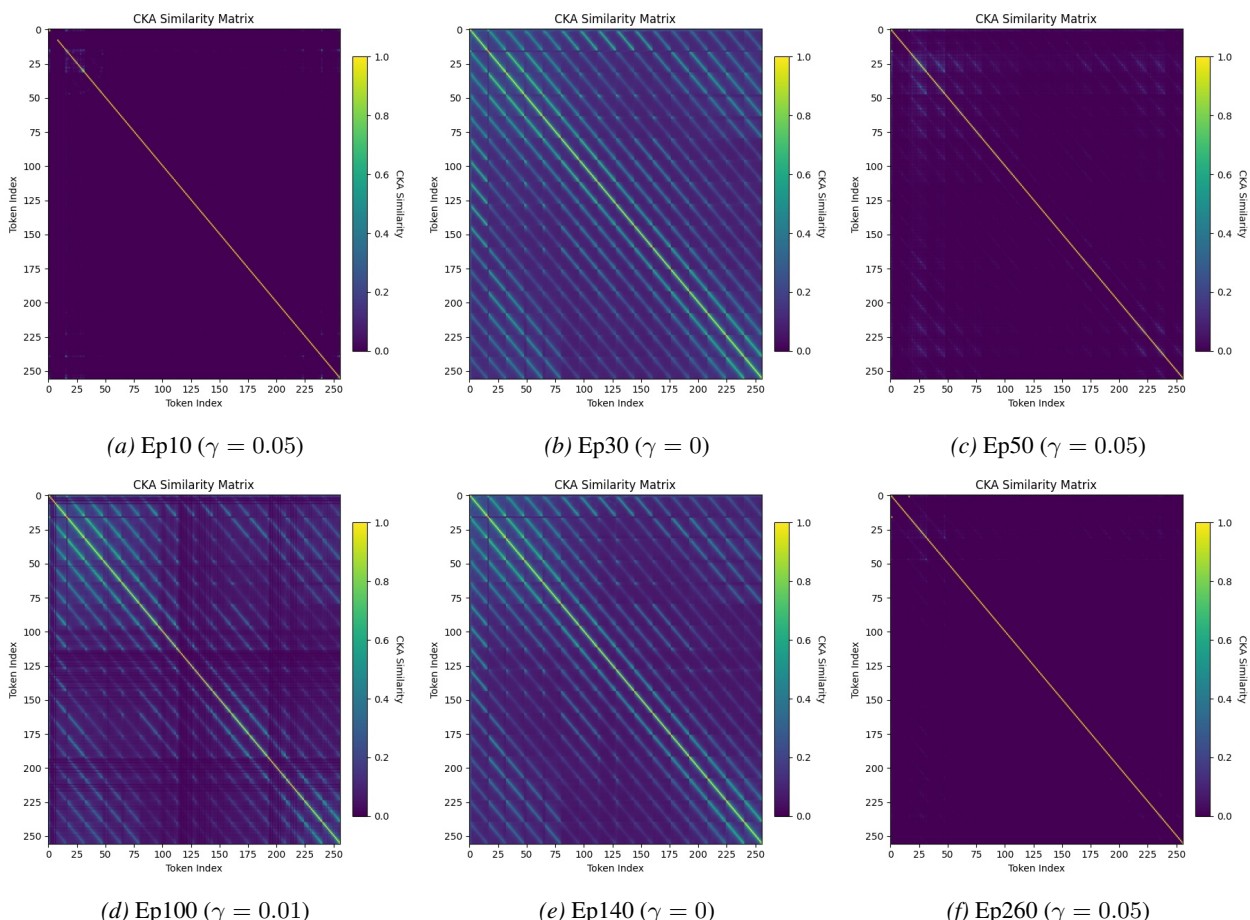

*Figure 19.* CKA plots for a single model at various points during training, while we vary the regularization weight $\gamma$.

## L.1 CKA Regularization Pseudocode

In algorithm 1 we show the pseudocode for computing the off-diagonal CKA regularization. Given that we are doing distributed training, and that different ranks will have a different sampled number of tokens, as well as different batch sizes, we need to deal with gathering data across the ranks. First, we find the minimum sampled $T$ across all ranks, and truncate the second dimension to match. This allows us to all-gather the tensors across the ranks, concatenating into $B$. Next, we stochastically subsample from $B$ and $T$ to reduce the tensor size. Each rank samples a different subset. Finally, each rank independently projects the dimension $C$ down to $d$ using a random projection. Finally, we compute $\text{CKA}_{\text{reg}}$ as defined in (9).

---

**Algorithm 1** Stochastic Linear CKA over Token Positions

---

**Require:** Input tensor $X \in \mathbb{R}^{B \times T \times C}$
**Require:** Projection dimension $d$, max batch size $B_{\max}$, max tokens $T_{\max}$
1: $T \leftarrow \min_{\text{ranks}} T$
2: Truncate $X \leftarrow X[:, 1:T, :]$
3: **if** $T < 2$ **then**
3:   **return** $0$
4: **end if**
5: All-gather $X$ across ranks along batch dimension
6: $B \leftarrow$ number of rows of $X$
7: **if** $B < 2$ **then**
7:   **return** $0$
8: **end if**
9: **if** $B > B_{\max}$ **then**
10:   Uniformly sample $B_{\max}$ batch indices
11:   Subsample batch dimension of $X$
12: **end if**
13: **if** $T > T_{\max}$ **then**
14:   Uniformly sample $T_{\max}$ token indices
15:   Subsample token dimension of $X$
16:   $T \leftarrow T_{\max}$
17: **end if**
18: Sample random projection $R \in \mathbb{R}^{C \times d}$ with

$$R_{ij} \sim \mathcal{N}\left(0, \frac{1}{d}\right)$$

19: Project features: $X \leftarrow XR \in \mathbb{R}^{B \times T \times d}$
20: Transpose: $X \leftarrow X^{\top} \in \mathbb{R}^{T \times B \times d}$
21: **for** $i = 1$ to $T$ **do**
22:   Mean-center token representations:

$$X_i \leftarrow X_i - \frac{1}{B}\sum_{b=1}^{B} X_i[b,:]$$

23: **end for**
24: **for** $i = 1$ to $T$ **do**
25:   Compute self Gram norm:

$$g_i \leftarrow \sqrt{\left\| X_i^{\top} X_i \right\|_F^2}$$

26: **end for**
27: Initialize $S \leftarrow 0$
28: **for** $i = 1$ to $T$ **do**
29:   **for** $j = 1$ to $T$ **do**
30:     **if** $i \neq j$ **then**
31:       Compute cross Gram Frobenius norm:

$$n_{ij} \leftarrow \left\| X_i^{\top} X_j \right\|_F^2$$

32:       Accumulate:

$$S \leftarrow S + \frac{n_{ij}}{g_i g_j}$$

33:     **end if**
34:   **end for**
35: **end for**
35: **return** $\dfrac{1}{T(T-1)} S$
   =0

---