# OpenReview forum: "RADIO1D: Elastic Representations for Condensed Vision Modeling"
_ICML.cc/2026/Conference — ICML 2026 regular_

### Official Review · Reviewer_Qi1e · 2026-03-10

**Soundness:** 2
**Presentation:** 2
**Significance:** 2
**Originality:** 2
**Overall Recommendation:** 3
**Confidence:** 3

**Summary:**

This paper proposes **RADIO1D**, which attempts to compress visually encoded tokens dynamically based on user queries. RADIO1D aims to provide flexible accuracy-efficiency tradeoffs through adjustable token counts.

**Compliance With Llm Reviewing Policy:**

Affirmed.

**Final Justification:**

As my concerns have been partially resolved, I raised my overall rating.

**Key Questions For Authors:**

Please check the weaknesses

**Limitations:**

No, the authors does not provide any limitation in this paper.

**Strengths And Weaknesses:**

# Strengths

The paper successfully combines the dynamic token compression mechanism used in **FlexTok** with the **C-RADIOv4** visual encoder to create a dynamic token compression framework for MLLMs.

# Weaknesses

1. **Poor Presentation:**
The paper is difficult to read, containing numerous run-on sentences, unexplained abbreviations, and missing citations. Examples include:
    * The following 90-word run-on sentence: *"Top VLMs adopting SigLIP include PaliGemma (Beyer et al., 2024), which integrates it for multimodal generation, Idefics2 (Laurençon et al., 2024), leveraging it for vision-language understanding with Llama-based decoders, Cambrian-1 (Tong et al., 2024), which employs an ensemble including SigLIP for vision-centric multimodal tasks, NVILA (Liu et al., 2024b) for high-resolution image and video processing, and Qwen3 VL (Bai et al., 2025), and RADIO (Heinrich et al., 2025) has been adopted in the Nemotron VL (Deshmukh et al., 2025) VLM family, for document intelligence, video understanding, and multi-image reasoning."*
    * Unexplained abbreviations such as **PDF** in Figure 2 and **HSIC** in Line 136.
    * Mention of **C-RADIOv2** in Line 76 without context or proper reference.

2. **Disorganized Content and Logical Flow:**
    * Given that the paper focuses on dynamic token compression based on user queries, the relationship between the proposed method and the "poor dense spatial coherence" of the visual encoder is unclear. The discussion of the C-RADIOv4 paradigm appears largely borrowed and unrelated to the actual dynamic compression mechanism proposed.
    * Section 3.4 devotes excessive space to describing a standard down-sampling operation and provides a theoretical complexity analysis that lacks novelty. These concepts have been extensively explored in prior work and are not directly related to the core dynamic compression mechanism.
    * The analysis regarding **register tokens** was previously established by Darcet et al. [1] two years ago. This work's analysis overlaps significantly with that prior research, yet it fails to provide citations or discussions, raising concerns regarding plagiarism or at least a lack of literature awareness.

3. **Insufficient Experimental Evaluation:**
The core objective of the paper is dynamic query-based compression; however, comparing performance only against baseline methods is insufficient. The authors should have compared their results with existing state-of-the-art (SOTA) visual token pruning methods. Furthermore, there is a total lack of discussion regarding existing visual token pruning literature.

4. **Marginal Acceleration vs. Significant Performance Drop:**
The actual acceleration gains appear minimal when compared to the resulting performance degradation. The authors claim that *"the resulting representations exhibit strong hierarchical summarization, enabling accurate scene understanding—even with a single token."* However, results in Table 1 show that using a single token achieves only roughly 50% of the baseline performance on OCR-related benchmarks such as **DocVQA** and **InfoVQA**.

---

# References

[1] Darcet, Timothée, et al. *"Vision transformers need registers."* arXiv preprint arXiv:2309.16588 (2023).

---

> ### Author Rebuttal · Authors · 2026-03-30
>
> # Presentation Issues
>
> We appreciate the reviewer highlighting presentation issues; we agree unexplained acronyms detract from readability. We will explicitly define terms upon first use (e.g., PDF: Probability Density Function, HSIC: Hilbert-Schmidt Independence Criterion) and break up run-on sentences.
> Regarding references, our strategy was to cite a work only upon its first mention to avoid redundancy, but we recognize this was applied inconsistently (e.g., C-RADIOv4 is introduced in Sec 2.1 but cited in Sec 2.2). We will review the manuscript to ensure all citations are correctly placed.
>
> # Darcet et al. (ICLR 2024)
>
> We agree we missed this reference, though it was an error of omission, not malice, since we used the exact terminology of “register.” We don’t think this meets the criteria of plagiarism, given we never tried to claim registers as our own invention; rather, we referenced the work by name while missing the citation. We will add a discussion of Darcet et al. to the “Related Work” to properly acknowledge this prior research. While both papers observe outlier tokens in trained ViTs, our methodologies, findings, and architectural goals are different:
>
> | Feature | Darcet et al. | RADIO1D (Ours) |
> | :--- | :--- | :--- |
> | **Identification** | Uses local attention maps and high feature norms. | Uses PCA feature maps, capturing both feature vector direction and norm. |
> | **Nature of Outliers** | Identifies high-norm "artifacts" or noise that degrade local 2D spatial representations. | Identifies low-norm tokens in image-text models (e.g., SigLIP2) that act as effective global summarizers. |
> | **Goal** | Preserves and cleans 2D spatial grid representations for dense prediction tasks. | Abandons the 2D grid for an elastic 1D sequence where tokens represent hierarchical levels of detail. |
> | **Intervention** | Appends new "register" tokens to absorb global noise and clean the spatial patch tokens. | Inherits registers from RADIOv4, but uses a 1D bottleneck to encode summaries rather than explicitly cleaning a 2D grid. |
>
> In summary, Darcet et al. identify artifacts that break 2D spatial representations and add registers to fix the grid. Our work observes that image-text aligned models naturally develop global summarizers, which we leverage to build a more efficient, elastic 1D sequence.
>
> # Comparison Against Visual Token Pruning
>
> We evaluated Token Merging (Bolya et al., ICLR 2023) on C-RADIOv4 at 128 and 192 tokens using a fully fine-tuned VLM. While efficient, ToMe showed lower accuracy than RADIO1D, particularly in dense OCR and table parsing. Examining failure cases in DocVQA, we observed the model frequently returning the wrong element from a table. This indicates that merging distant tokens degrades spatial awareness, causing the model to identify correct elements but lose their spatial grounding.
>
> | Encoder | Tokens | TextVQA | DocVQA | InfoVQA | OCRBench | OCRBv2-EN | OCRBv2-CN | AI2D | ChartQA | MMMU | SeedB | LVBench | Avg |
> | :--- | :--- | :--- | :--- | :--- | :--- | :--- | :--- | :--- | :--- | :--- | :--- | :--- | :--- |
> | RADIO1D | 128 | **84.0** | **92.5** | **72.6** | **82.8** | **61.5** | **38.6** | **85.1** | **88.6** | 47.8 | **77.6** | 57.0 | **71.64** |
> | C-RADIOv4+ToMe | 128 | 82.2 | 90.8 | 69.0 | 81.4 | 58.3 | 36.3 | 84.7 | 87.7 | **49.2** | 76.7 | **57.2** | 70.31 |
> | | | | | | | | | | | | | | |
> | RADIO1D | 192 | **84.5** | **93.2** | **75.6** | **83.5** | **60.9** | **40.2** | **85.7** | **89.2** | 48.2 | **77.6** | **57.3** | **72.36** |
> | C-RADIOv4+ToMe | 192 | 83.7 | 91.7 | 71.9 | 82.9 | 59.6 | 40.1 | 84.5 | 88.5 | **49.8** | 77.3 | 56.4 | 71.49 |
>
> # Effective Acceleration in VLM Context
>
> Regarding the observed speedups (TTFT), we chose to report actual system measurements using vLLM rather than theoretical FLOP reductions, as we believe this better reflects real-world deployment. However, it is important to note that the vision inference path in current serving frameworks has known bottlenecks. Specifically, image pre-processing takes roughly as much time as the actual vision model inference (documented [here](https://github.com/vllm-project/vllm/issues/20799)). Consequently, these system-level, CPU-bound inefficiencies partially mask the true architectural speedup of RADIO1D. We prioritized reporting authentic, reproducible measurements, even though they understate the theoretical efficiency gains of our 1D sequence design.

---

> > ### Author Rebuttal · Reviewer_Qi1e · 2026-04-04
> >
> > I thank the authors for their detailed response, and I am glad to see that the presentation issues will be revised. However, the "Comparison Against Visual Token Pruning" should be conducted against token pruning methods designed specifically for MLLMs (e.g., CDPruner [1]), rather than those for the vision backbone, to demonstrate the true effectiveness of the proposed framework.
> >
> > As my concerns have been partially resolved, I am inclined to raise my rating. I will make my final decision after the reviewer discussion stage.
> >
> > [1] Beyond Attention or Similarity: Maximizing Conditional Diversity for Token Pruning in MLLMs

---

> > > ### Author Response · Authors · 2026-04-06
> > >
> > > This is a fantastic suggestion, thank you!
> > > We integrated CDPruner into our test bench using the greedy Determinantal Point Process (DPP) algorithm from their [repository](https://github.com/Theia-4869/CDPruner).
> > > For the text query embedding, we average-pooled the token embeddings from the language model's embedding lookup table.
> > > For the vision tokens, we used the output of the vision projector, so both are in the same embedding space.
> > > We then applied greedy DPP as a training-free token selection method, independently on each image tile.
> > > Since CDPruner is orthogonal to RADIO1D, we experimented with it on top of both C-RADIOv4 (2D) tokens and RADIO1D (full 256 tokens).
> > > Notably, RADIO1D at 128 tokens (71.64 avg) outperforms all CDPruner configurations at 128 tokens and even surpasses C-RADIOv4+ToMe at 192 tokens (71.49). At 192 tokens the gap narrows, but RADIO1D remains the best overall. We include the full 256-token baselines for reference.
> > >
> > >
> > > | Encoder | Tokens | TextVQA | DocVQA | InfoVQA | OCRBench | OCRBv2-EN | OCRBv2-CN | AI2D | ChartQA | MMMU | SeedB | LVBench | Avg |
> > > | :--- | :--- | :--- | :--- | :--- | :--- | :--- | :--- | :--- | :--- | :--- | :--- | :--- | :--- |
> > > | **C-RADIOv4+ToMe** | 128 | 82.2 | 90.8 | 69.0 | 81.4 | 58.3 | 36.3 | 84.7 | 87.7 | 49.2 | 76.7 | 57.2 | 70.31 |
> > > | **C-RADIOv4+CDPruner** | 128 | 82.8 | 89.2 | 69.2 | 74.7 | 52.9 | 33.9 | 83.5 | 87.6 | 50.3 | 77.0 | **58.4** | 69.05 |
> > > | **RADIO1D(256 tokens)+CDPruner** | 128 | 82.2 | 87.2 | 67.6 | 76.5 | 54.6 | 31.9 | 84.2 | 86.8 | **50.4** | 76.2 | 56.3 | 68.56 |
> > > | **RADIO1D (ours)** | 128 | **84.0** | **92.5** | **72.6** | **82.8** | **61.5** | **38.6** | **85.1** | **88.6** | 47.8 | **77.6** | 57.0 | **71.64** |
> > > | | | | | | | | | | | | | | |
> > > | **C-RADIOv4+ToMe** | 192 | 83.7 | 91.7 | 71.9 | 82.9 | 59.6 | 40.1 | 84.5 | 88.5 | 49.8 | 77.3 | 56.4 | 71.49 |
> > > | **C-RADIOv4+CDPruner** | 192 | 84.1 | 92.5 | **75.8** | 82.1 | 58.2 | 39.4 | 85.3 | **89.4** | **51.1** | **77.8** | **58.9** | 72.23 |
> > > | **RADIO1D(256 tokens)+CDPruner** | 192 | 83.8 | 92.3 | 75.3 | 82.2 | 59.7 | 39.4 | **86.0** | 88.7 | 50.6 | 77.2 | 57.1 | 72.02 |
> > > | **RADIO1D (ours)** | 192 | **84.5** | **93.2** | 75.6 | **83.5** | **60.9** | **40.2** | 85.7 | 89.2 | 48.2 | 77.6 | 57.3 | **72.36** |
> > > | | | | | | | | | | | | | | |
> > > | **C-RADIOv4** | 256 | **84.3** | **93.3** | 77.7 | 84.3 | 60.4 | 41.5 | 86.0 | 89.1 | **50.8** | **78.1** | **58.6** | 73.10 |
> > > | **RADIO1D (ours)** | 256 | 84.2 | 93.2 | **78.6** | **84.4** | **61.4** | **42.2** | **86.9** | 89.1 | 50.7 | 77.9 | 57.8 | **73.31** |
> > >
> > >
> > > CDPruner is a training-free method applied post-hoc, while RADIO1D learns to compress tokens during training. The results suggest that learning what to preserve at the backbone level is more effective than selecting tokens after the fact.
> > > We also found that CDPruner struggles with dense OCR tasks, which is consistent with the OCRBench results reported in their paper.
> > >
> > >
> > > As you suggested, we will shorten the paper section about downscaling block architecture in favor of expanding the section on related work and baselines.
> > >
> > > We are very grateful for the valuable feedback, which definitely helped improve the manuscript.

---

### Official Review · Reviewer_YGLG · 2026-03-12

**Soundness:** 3
**Presentation:** 2
**Significance:** 2
**Originality:** 3
**Overall Recommendation:** 4
**Confidence:** 3

**Summary:**

This paper challenges the assumption that VLMs require fixed 2D patch-based visual features, proposing instead to compress image representations into variable-length 1D token sequences for flexible accuracy-efficiency tradeoffs. The paper proposes RADIO1D employs multi-teacher knowledge distillation (SigLIP2-g, DINOv3-7B, SAM3) combined with integrated Patch Merging and nested dropout, enabling a single checkpoint to produce 1–256 hierarchically ordered tokens. It uses a lightweight autoencoder decoder reconstructs 1D tokens back to a 2D grid for teacher alignment during training only.
Experiments on 10 multimodal benchmarks and 9B Nemotron LLM show that RADIO1D at 256 tokens
  matches the performance of fixed-length 2D baselines like C-RADIOv4-H, while remaining competitive under aggressive compression (e.g., 70.07% average accuracy at 64 tokens).

**Compliance With Llm Reviewing Policy:**

Affirmed.

**Final Justification:**

The rebuttal addresses my concens.

**Key Questions For Authors:**

1. Per-image token allocation. Have you analyzed whether there exist image subsets where
   using fewer tokens yields no performance loss compared to the full 256 tokens? If so,
  what proportion of images fall into this category? A positive result would substantially
   strengthen the motivation for elastic representations and address my concern in W2.
  2. Comparison with token pruning. How does RADIO1D compare against representative token
  pruning methods (e.g., FastV, ToMe) applied to C-RADIOv4-H at matched token budgets on
  the same VLM benchmarks? If RADIO1D shows clear advantages at low token counts, this
  would better justify the need for a dedicated 1D encoder over post-hoc pruning.
  3. Disentangling the source of gains. Since RADIO1D is initialized from a pretrained
  C-RADIOv4 checkpoint and further fine-tuned with multi-teacher distillation, how much of
   the performance comes from the 1D elastic design versus the distillation training
  itself? Specifically, if the 2D features before 1D conversion are directly fed into the
  LLM, what is the performance gap compared to RADIO1D? This would help clarify whether
  the 1D formulation provides meaningful benefits beyond what the multi-teacher training
  already offers.

**Limitations:**

yes

**Strengths And Weaknesses:**

**Strengths**
- S1. Well-grounded motivation. The paper demonstrates through empirical
  analysis (CKA matrices, PCA visualization) that while pretrained vision encoders exhibit
   spatial coherence, this spatial structure is largely discarded during VLM fine-tuning.
  It shows that VLMs do not fundamentally rely on 2D spatial positioning. It provides
  a well-justified motivation for exploring 1D visual representations.

- S2. Comprehensive experimental evaluation. The paper provides thorough empirical
  validation across multiple dimensions, including preliminary analyses of vision encoder
  features, ablation studies on key design choices, rate-distortion analysis, VLM
  performance comparison across 10 multimodal benchmarks, and composition-based retrieval
  evaluation. Together, these experiments sufficiently demonstrate the effectiveness of
  the proposed method.

- S3. Pareto-optimal accuracy-latency tradeoff. Figure 1 demonstrates that
  RADIO1D achieves the Pareto frontier on VLM accuracy vs. time-to-first-token, offering a
   continuous spectrum of operating points from a single model. At 128 tokens, RADIO1D
  already matches C-RADIOv4-H's average accuracy (72.36% vs 73.09%) while being
  substantially faster, which is practically meaningful for deployment.

**Weaknesses**
- W1. Limited methodological novelty. The core approach largely transfers FlexTok's
  elastic tokenization mechanism to visual representation learning, with the main
  modification being replacing the reconstruction target with multi-teacher distillation.
  The individual components (nested dropout, hierarchical encoding, autoencoder decoder)
  are drawn from existing work, and the paper does not sufficiently articulate why this
  adaptation is non-trivial.

 - W2. Flexible token length advantage not convincingly demonstrated. The experimental
  results consistently show that more tokens monotonically lead to better performance
  (Table 1), with no analysis of whether certain images can be adequately represented with
   fewer tokens. A more compelling demonstration would show dynamic per-image token
  allocation based on content complexity. Without such evidence, the flexible length
  property simply reduces to using fewer tokens for worse performance, diminishing the
  practical significance of the elastic design.

 - W3. Missing comparison with token pruning methods. Existing inference-time token pruning
   methods (e.g., FastV, ToMe) can also reduce visual tokens while maintaining competitive
   VLM performance. The paper lacks any comparison at matched token budgets. This is
  particularly concerning given that RADIO1D shows notable degradation at low token counts
   (e.g., 63.88% at 8 tokens vs. 73.29% at 256), making it unclear whether a specialized
  1D encoder offers advantages over simply pruning tokens from a strong 2D encoder.

---

> ### Author Rebuttal · Authors · 2026-03-30
>
> # Novelty of the Work
>
> Our technical novelty lies in combining analytical insights with an architectural design tailored for VLMs:
> * **Analytical Insight:** "Artifact" tokens in models like SigLIP2 are highly effective global summarizers, not spatial noise. This motivated our shift from 2D grids to elastic 1D sequences.
> * **Semantic Reconstruction:** Unlike FlexTok (which reconstructs raw pixels), RADIO1D learns by distilling diverse semantic features from multiple teachers. This agglomerative training makes the tokens useful for complex reasoning.
> * **Architectural Engineering:** Adapting 1D sequences for VLMs is not plug-and-play. Because 1D tokens lack a 2D layout, standard operations (like 2x2 unshuffling) fail. We engineered a custom 1D-encoder/2D-decoder bridge for teacher alignment and introduced a learnable 1D patch merging mechanism.
>
> # Content-Aware Token Allocation
>
> We are glad you brought up the fascinating concept of content-aware compression. We have spent a lot of time pondering exactly this: given an image, is it feasible to dynamically predict the "perfect" number of tokens to represent it?
>
> While image complexity plays a role, our hypothesis is that the task, rather than the image itself, is the primary driving factor for token requirements. Using too many tokens wastes compute, while using too few falls short of the specific task's demands. To illustrate this, we pivoted the results from Table 1 (and our ADE20k evaluation) to determine the minimum number of tokens required to reach within 1%, 5%, and 10% of the peak score for various benchmarks:
>
> | Category | Benchmark | Min. Tokens (w/in 10%) | Min. Tokens (w/in 5%) | Min. Tokens (w/in 1%) |
> |:---|:---|:---|:---|:---|
> | Dense Spatial | ADE20k | 4 | 8 | 32 |
> | Dense OCR | InfoVQA | 128 | 192 | 256 |
> | | OCRBench | 32 | 64 | 224 |
> | | OCRBenchv2-EN | 32 | 64 | 128 |
> | | OCRBenchv2-CN | 128 | 224 | 224 |
> | Document/Chart | DocVQA | 32 | 64 | 128 |
> | | ChartQA | 8 | 32 | 128 |
> | | TextVQA | 8 | 32 | 128 |
> | | AI2D | 8 | 32 | 224 |
> | Reasoning & Video | LVBench | 8 | 32 | 192 |
> | | SeedBench | 8 | 8 | 128 |
> | | MMMU | 1 | 224 | 224 |
>
> *Note: The peak score is the maximum score achieved by any RADIO1D configuration for that benchmark, and thresholds are relative %.*
>
> As the data shows, just 32 tokens are sufficient to achieve peak mIoU on the ADE20k corpus. Conversely, reading-heavy OCR tasks almost strictly require the full 256-token sequence to resolve fine text. Similarly, video understanding requires far fewer tokens per frame to maintain strong performance. If the task was nearly lossless image reconstruction, we would naturally need many more tokens.
>
> We don't want to overstate this hypothesis, but the data suggests that the task itself determines the best token count. This highlights the real-world value of our elastic design: RADIO1D gives its users a single model that can be adjusted to provide the exact number of tokens needed for a specific job without retraining the vision encoder.
>
> # Token Pruning Baselines
>
> As detailed in our response to Reviewer Qi1e, we evaluated Token Merging (Bolya et al., ICLR 2023) on C-RADIOv4 at 128 and 192 tokens. While efficient, ToMe showed lower accuracy than RADIO1D, particularly in dense OCR and table parsing. In fact, we examined failure cases in DocVQA and noticed the model frequently responded with the wrong element of a table. This indicates that merging distant tokens degrades spatial awareness, leading the model to identify the correct elements but in the wrong locations.
>
> Regarding FastV, it is important to note that FastV and RADIO1D are complementary. FastV dynamically prunes tokens inside the LLM during decoding, whereas RADIO1D compresses the visual sequence before it reaches the LLM. A system can apply both in parallel for maximum efficiency. Furthermore, FastV is LLM-specific, whereas RADIO1D's compressed representations improve efficiency across non-VLM applications (e.g., retrieval, dense segmentation).
>
> # Comparison against RADIO (2D)
>
> We apologize if this was not clear enough in the paper. The comparison you asked for (testing the 2D features before our 1D conversion and distillation) is actually our C-RADIOv4-H baseline. To ensure a fair comparison, the C-RADIOv4-H 2D baseline starts with 1024 spatial tokens, but we apply a standard 2x2 pixel unshuffle to reduce it to exactly 256 tokens before feeding it to the LLM. Therefore, both models give the LLM exactly 256 tokens. As shown in Table 1:
> * **C-RADIOv4-H (2D, unshuffled to 256 tokens):** 73.09% average accuracy | 468.2 ms TTFT
> * **RADIO1D (1D sequence, 256 tokens):** 73.29% average accuracy | 452.8 ms TTFT
>
> At the same token budget, the 1D sequence is more accurate on average and processes faster. Freed from a rigid 2D grid, 1D tokens naturally align better with the LLM's 1D format.

---

> > ### Author Rebuttal · Reviewer_YGLG · 2026-04-02
> >
> > The rebuttal addresses my main concerns well. The token pruning comparison, the C-RADIOv4-H baseline clarification, and the per-benchmark minimum token analysis are convincing. I am raising my score to 4.
> >
> > One discussion point on the Content-Aware Token Allocation analysis: I agree that the required token count is task-dependent. However, for a given image, there should exist an information-theoretic upper bound a minimum number of tokens sufficient for any task. Task-specific requirements would then be a lower bound of this. Have you considered analyzing this image-level upper bound? If it is significantly below 256 for many images, it would further strengthen the practical value of elastic design.

---

> > > ### Author Response · Authors · 2026-04-06
> > >
> > > We completely agree that for any given image, there should exist an information-theoretic upper bound of necessary tokens, for example, successful lossless compression (e.g. PNG) demonstrates that not all bits are necessary. The existence of this image-level bound is what motivates the development of elastic architectures like RADIO1D, as our method provides the required mechanism to actually scale down to that limit.
> > >
> > > However, efficiently predicting this optimal image-specific bound on the fly is a complex problem indeed, since leveraging methods like InfoTok [2] would require running inference for all of the teachers in the RADIO regime, and is not well defined within the VLM after the vision model has been finetuned in any way. We are actively investigating lightweight predictor networks to achieve this image-aware token allocation as part of our ongoing research.
> > >
> > > Thank you very much for the insightful comments!
> > >
> > > [2] InfoTok: Adaptive Discrete Video Tokenizer via Information-Theoretic Compression (https://openreview.net/forum?id=JEYWpFGzvn)

---

### Official Review · Reviewer_VPbA · 2026-03-13

**Soundness:** 3
**Presentation:** 3
**Significance:** 2
**Originality:** 2
**Overall Recommendation:** 4
**Confidence:** 3

**Summary:**

The core goal is to develop a more efficient image representation that still preserves rich semantic structure for downstream vision-language tasks. They address this by introducing RADIO1D, a method that compresses images into a variable-length 1D token sequence, in contrast to traditional large patch-based encodings.

Their research question focuses on whether this compact representation can both improve alignment with language and reduce computational cost. The novelty lies in their flexible, elastic tokenization, which balances efficiency and semantic coherence. Compared to prior work, they retain spatial structure better, enabling improved downstream reasoning.

Their contributions include (1) introducing the RADIO1D image tokenization method, (2) demonstrating its effectiveness on standard vision-language benchmarks, (3) showing a significant efficiency boost in terms of FLOPs and memory usage, and (4) preserving spatial coherence in their learned tokens.

They fine-tune the CLIP and SigLIP image encoder on the MS COCO dataset for image captioning, and they evaluate it on the VQAv2 dataset for visual question answering. Their main evaluation metric is accuracy on these tasks, and they show that, despite the reduced computational cost, their model achieves competitive or superior performance, striking a balance between efficiency and accuracy.

**Compliance With Llm Reviewing Policy:**

Affirmed.

**Key Questions For Authors:**

Q1. The paper’s core analysis in Section 2 centers mainly on SigLIP2 and DINOv3; could the authors show whether the same conclusions about spatial coherence, global tokens, and post-VLM fine-tuning degradation also hold for other widely used encoders such as CLIP, EVA-CLIP, or DeiT vision backbones?


Q2. The paper argues that VLM fine-tuning shifts representations toward more abstract but less spatially coherent features, yet it is still not fully clear which part of the VLM training pipeline causes this effect most strongly; can the authors disentangle whether this comes primarily from the projector, the language supervision, or full end-to-end supervised fine-tuning?


Q3. RADIO1D shows strong results with very few tokens, sometimes even a single token, but the paper does not fully explain the failure modes of this compression regime; could the authors provide a more detailed analysis of which task types or image characteristics break down first as the token budget becomes extremely small?

I can also turn these into shorter, sharper reviewer-style questions.

**Limitations:**

The paper gives a limited analysis of failure cases on harder reasoning-heavy multimodal tasks, so it is still unclear when aggressive token compression starts to hurt performance. A useful improvement would be stronger task-level error analysis on OCR, localization, and multi-step reasoning benchmarks.


The study is validated on only a small set of vision encoders, which makes the generality of its conclusions less certain. A clear improvement would be to test the method and analysis on more encoder families, such as CLIP, EVA-CLIP, or DeiT backbones.

**Strengths And Weaknesses:**

**Strengths**

The method is conceptually strong because it introduces a variable-length 1D visual representation that gives a clear accuracy–efficiency tradeoff, rather than forcing a fixed and often redundant 2D patch grid.

The experiments are broad and convincing in scope, covering representation analysis, ablations, composition-aware retrieval, semantic segmentation behavior, and full VLM evaluation across 10 multimodal benchmarks, which makes the empirical case much stronger than a narrow benchmark-only study.

The paper does a good job connecting analysis to method design: the observations about spatial coherence, global summarization tokens, and the effect of VLM fine-tuning directly motivate RADIO1D’s architecture and training strategy.


**Weaknesses**

Although the method is well motivated, the practical impact may feel somewhat limited because it mainly improves the vision encoder side of the pipeline; this is still valuable, but the field’s center of gravity has increasingly shifted toward end-to-end multimodal large language models, where system-level reasoning, instruction tuning, and multimodal alignment are often the main focus.


The experimental story is strong on efficiency and compression, but it is less clear how robust the method is on harder reasoning-heavy multimodal settings, since most evidence emphasizes representation quality, retrieval, and benchmark averages more than deeper failure analysis or fine-grained reasoning behavior.

---

> ### Author Rebuttal · Authors · 2026-03-29
>
> # Criticality of the Vision Encoder
>
> We agree with the reviewer that the field's focus is shifting toward end-to-end MLLMs and system-level reasoning. We actually view highly efficient vision encoders as an important enabler for these system-level goals. As noted in our introduction, frontier models all rely on pre-trained vision backbones. Because an LLM's reasoning capacity is strictly bottlenecked by its context window, visual token efficiency is a system-wide constraint.
> By compressing images into a much more compact 1D sequence, RADIO1D empowers these end-to-end models to process reasoning-heavy contexts with less computational overhead. We show in Table 1 how RADIO1D enables negligible drops in reasoning-heavy MMMU with a fraction of the initial token count.
>
> # Q1: Generalization to other vision encoders
>
> We focused specifically on SigLIP2 and DINOv3 because they represent the state-of-the-art extremes for image-text alignment and dense spatial representation, respectively. In our experience, other contrastive models like OpenAI CLIP and DFN CLIP exhibit very similar behaviors and VLM performance characteristics to SigLIP2. We chose SigLIP2 simply to demonstrate this phenomenon using the strongest possible baseline currently available.
>
> # Q2: Disentangling Feature Adaptation
>
> In our training pipeline, the vision encoder remains completely frozen during the initial vision-language alignment phase. The projector is simply a 2-layer MLP that processes every token independently and identically; it has no spatial awareness beyond what it inherits from the vision encoder's positional encodings. Therefore, the spatial structure of the visual features is only altered later, when the vision encoder is finally unfrozen and updated using language supervision during SFT.
>
> # Q3: Failure Modes at Extreme Compression
>
> To identify these failure modes, we analyzed the data in Table 1. We mapped out exactly when performance breaks down for different tasks (also detailed in our response to Reviewer YGLG).
> We found that tasks requiring holistic scene understanding to handle extreme compression very well. These include video reasoning, semantic segmentation, and even complex reasoning like MMMU. In fact, MMMU stays within 10% of its peak score using just a single token.
> Conversely, the primary failure mode occurs in dense, reading-heavy tasks. Performance on OCR and document QA drops sharply as the token count decreases. We will add this task-specific analysis to the manuscript to clearly define the limits of extreme compression.

---

> > ### Author Rebuttal · Reviewer_VPbA · 2026-04-04
> >
> > I appreciate the author's responses. They cover most of my concerns. Yet, I stick to my score, also considering the mentioned points from the Reviewer Qi1e that I think are really important and show some weaknesses in the evaluation I formerly did not notice.

---

> > > ### Author Response · Authors · 2026-04-06
> > >
> > > Thank you for your continued engagement!
> > >
> > > We understand your reservation regarding the evaluation gaps raised by Reviewer Qi1e.
> > > To directly address this, we have now conducted the requested evaluation against CDPruner, a state-of-the-art MLLM-specific token pruning method. As detailed in our latest response to Reviewer Qi1e, the empirical results show:
> > >
> > > * Superior Performance: RADIO1D outperforms CDPruner at equivalent token budgets (e.g., 71.64 vs 69.05 average accuracy at 128 tokens).
> > > * Task Robustness: RADIO1D effectively retains crucial spatial information for dense OCR tasks, whereas post-hoc pruning methods like CDPruner struggle.
> > >
> > > These results confirm that learning compression during vision backbone training is more effective than post-hoc token selection. We invite you to review the full data table in our response to Reviewer Qi1e, and we hope this resolves your remaining concerns regarding the evaluation.

---

### Official Review · Reviewer_rdu2 · 2026-03-18

**Soundness:** 3
**Presentation:** 3
**Significance:** 4
**Originality:** 4
**Overall Recommendation:** 5
**Confidence:** 4

**Summary:**

This paper proposes RADIO1D that can faithfully distort the regular grid-like image patch structure into an elastic 1D sequence with the the first tokens carry global and semantic information and latter tokens carry details. It adopts RADIOv4 training recipe but add an encoder decoder architecture to perform 2D -> 1D construction and 1D -> 2D restoration. The experiment are thoroughly conducted and results are convincing.

**Compliance With Llm Reviewing Policy:**

Affirmed.

**Key Questions For Authors:**

No.

**Limitations:**

yes.

**Strengths And Weaknesses:**

Strengths: This paper explores the possibility of modeling image by an elastic 1D representation instead of 2D grid structure via agglomerative training, the analysis in Section 2.1 and 2.2 are insightful to motivate this method. The experiments are comprehensive spanning from multi-modal understanding to traditional perception tasks. The performance is also convincing due to the already successful RADIOv4 training recipe. I believe this is a foundational work that enables fewer image tokens and eases the visual computational burden of current MLLMs.

---

> ### Author Rebuttal · Authors · 2026-03-29
>
> We thank the reviewer for recognizing the potential of RADIO1D's elastic representations.
> To further strengthen the camera-ready manuscript, we are incorporating constructive feedback from the broader review panel. This includes adding comparisons against inference-time token pruning baselines (e.g., ToMe) to validate our efficiency advantages, as well as revising the text to improve the overall presentation.

---

### Decision · Program_Chairs · 2026-04-30

**Decision:**

Accept (regular)

**Comment:**

Most reviewers are positive about the paper’s empirical strength and practical value, while noting some limits in novelty and analysis. One reviewer raises a  concern that the novelty is incremental and that the initial evaluation lacked stronger multimodal token-pruning baselines.
The rebuttal partially addresses this by clarifying and the follow-up further strengthens the case with more pruning-baseline comparison. Overall, the concern is valid and mostly addressed on the evaluation side, while the novelty concern remains only partially resolved.